# Rapid localized spread and immunologic containment define Herpes simplex virus-2 reactivation in the human genital tract

Joshua T Schiffer[1,2]*, David Swan[1], Ramzi Al Sallaq[1†a], Amalia Magaret[1,3], Christine Johnston[1,4], Karen E Mark[4†b], Stacy Selke[3], Negusse Ocbamichael[3], Steve Kuntz[3], Jia Zhu[1,3], Barry Robinson[3], Meei-Li Huang[3], Keith R Jerome[3], Anna Wald[1,3,4,5], Lawrence Corey[1,3,4]

[1]Vaccine and Infectious Diseases Division, Fred Hutchinson Cancer Research Center, Seattle, United States; [2]Department of Medicine, University of Washington, Seattle, United States; [3]Department of Laboratory Medicine, University of Washington, Seattle, United States; [4]Department of Medicine, University of Washington, Seattle, United States; [5]Department of Epidemiology, University of Washington, Seattle, United States

*For correspondence: jschiffe@fhcrc.org

†Present address: [a]College of Nursing Global, New York University, New York, United States; [b]The Office of AIDS, California Department of Public Health, Sacramento, United States

Competing interests: The authors declare that no competing interests exist.

**Abstract** Herpes simplex virus-2 (HSV-2) is shed episodically, leading to occasional genital ulcers and efficient transmission. The biology explaining highly variable shedding patterns, in an infected person over time, is poorly understood. We sampled the genital tract for HSV DNA at several time intervals and concurrently at multiple sites, and derived a spatial mathematical model to characterize dynamics of HSV-2 reactivation. The model reproduced heterogeneity in shedding episode duration and viral production, and predicted rapid early viral expansion, rapid late decay, and wide spatial dispersion of HSV replication during episodes. In simulations, HSV-2 spread locally within single ulcers to thousands of epithelial cells in <12 hr, but host immune responses eliminated infected cells in <24 hr; secondary ulcers formed following spatial propagation of cell-free HSV-2, allowing for episode prolongation. We conclude that HSV-2 infection is characterized by extremely rapid virological growth and containment at multiple contemporaneous sites within genital epithelium.

## Introduction

The dynamics of most chronic viral infections such as HIV, hepatitis B and C, and cytomegalovirus are assessed by serially sampling blood using polymerase chain reaction (*Perelson et al., 1996*), and by ex vivo quantitation of circulating lymphocytes to estimate immune response (*Sylwester et al., 2005*). For disseminated infections, sampling from blood may reflect true host–pathogen dynamics due to homogeneous mixing of viruses and PBMCs, which originate from thousands of infectious foci. However, many infections are confined to small anatomic regions; localized host–pathogen interactions within specific organ tissue constitutes the predominant site of pathogenesis, and measurements of viral replication and immune response are not accurately reflected in the blood compartment. Moreover, sampling of infected tissue often reveals that the density of virus and host immune cells differ enormously across small distances. Spatial features of infection are a critical, and relatively unexplored, component of viral dynamics.

Herpes simplex virus-2 (HSV-2) is useful for studying localized aspects of pathogenesis in humans. HSV-2 replicates in highly accessible genital keratinocytes. Serial genital swabbing reveals frequent, highly variable shedding episodes in most infected persons (*Wald et al., 1995*, *1997*, *2000*;

**eLife digest** Viruses infect organisms as diverse as unicellular bacteria, plants, and animals. Two well-known human viral infections are herpes simplex virus 1, which is responsible for most cold sores, and herpes simplex virus 2 (HSV-2), which causes most cases of genital herpes. The first signs of HSV-2 infection are genital lesions, which usually heal relatively quickly. However, the virus can also enter nerve cells, where it hides from the immune system and survives for the lifetime of the infected host. In a process that is not fully understood, the dormant viruses inside nerve cells periodically reactivate and are transported back to the genital tract where they can cause recurrent genital sores. HSV-2 also commonly replicates in genital skin when lesions are not present, and is efficiently transmitted to other individuals at these times.

Most studies of viral infection rely on the examination of blood to determine viral load and characterize the immune response. However, the true battle between HSV-2 and its host occurs within genital tissues. HSV-2 lesions are clearly visible and it is therefore possible to precisely follow the number of viruses and the intensity of the host immune response as a function of time and position. Schiffer et al. follow this approach by examining the levels of HSV-2 DNA in the genital tracts of over 600 individuals, who were sampled at different time intervals and at multiple sites. Dramatic differences in the viral load and the density of CD8+ T cells, which are critical for controlling HSV-2, are observed across several millimeters of the skin. Viral load also varies dramatically over a few hours. These data are used to develop a mathematical model that characterizes the spatial dynamics of HSV-2 reactivation and suggests how the host immune system responds to reactivation.

Historically, it was thought that host immune cells, including CD8+ T cells, required many days to contain an HSV-2 reactivation, allowing prolonged symptoms. However, the model developed by Schiffer et al. indicates that although viral infection can spread from a single skin cell to thousands of cells in 12 hr or so, the host immune cells typically clear all infected skin cells in under 24 hr. Episodes of viral shedding lasting >3 days are predicted to occur due to viral seeding of adjacent regions of genital skin.

Schiffer et al. predict that the complex episodic nature of HSV-2 reactivation is heavily influenced by the spatial distribution and density of immune cells within the genital tract, which means that new lesions occur in regions where immune cell density is low. The rapid onset and localized clearance pattern suggests that viral elimination by skin-resident CD8+ T cells is actually highly effective within single genital ulcers. This hypothesis represents an area of intense current investigation for clinicians, virologists, mathematical modelers, and immunologists.

---

Crespi et al., 2007; Mark et al., 2008), and biopsies show dense clusters of CD4+ and CD8+ lymphocytes in focal areas of viral replication (Zhu et al., 2007). Some episodes are associated with lesions, and prolonged viral production, while others are asymptomatic and brief (Mark et al., 2008). Markedly different episodes occur within 5–10 days of each other, and viral levels fluctuate dramatically even within a single episode (Mark et al., 2008; Schiffer et al., 2011). At least 20% of episodes are notable for complex erratic viral trajectories, including re-expansion phases that represent failure of the immune system to contain replication (Schiffer et al., 2011). Past modeling suggests that nearly constant reactivation of HSV in a small percentage of infected ganglionic neurons might account for high frequency of shedding episodes (Schiffer et al., 2009). Animal and mathematical models predict that varying T-cell responses account for episode heterogeneity (Gebhardt et al., 2009; Schiffer et al., 2010). However, the field lacks hypotheses to address the complex morphology of prolonged viral shedding episodes.

We performed detailed studies with genital sampling performed over four different time intervals, and concurrently at multiple sites across the genital tract, to precisely define spatiotemporal kinetics of genital HSV-2 replication in immunocompetent patients. We then designed a mathematical model as a tool to develop hypotheses that explain virological data from these cohorts. Model simulations suggest very rapid rates of HSV-2 replication and epidermal cell-to-cell spread. Yet, host containment of virus within a single focus of infection occurs in <24 hr, necessitating seeding of adjacent regions to promote increased shedding.

## Results

### Temporal kinetics of HSV-2 genital replication

HSV-2 shedding in the genital tract is characterized by intermittent frequent episodes of variable duration (hours to weeks) and viral load (*Mark et al., 2008*; *Schiffer et al., 2011*). To evaluate viral patterns during individual shedding episodes, we analyzed data from three cohorts of immunocompetent patients who underwent genital tract swabbing at different time intervals (*Table 1*; *Boxes 1–3*). The same collection methods and HSV PCR assay were used in each cohort. Swabs were tested for HSV DNA using a validated quantitative PCR assay with a sensitivity of 1 copy/reaction (*Jerome et al., 2002*; *Magaret et al., 2007*). These studies demonstrated that quantity of genital HSV DNA is stable over minutes but expands and decays extremely rapidly over hours, and that prolonged episodes are notable for frequent and erratic peaks in HSV-2 viral load (*Figure 1*, *Figure 1—figure supplements 1–3*).

### Spatial features of HSV-2 replication and immunological response

Based on the observation that HSV-2 reactivation occurs simultaneously in multiple regions across the genital tract (*Tata et al., 2010*), we designed a fourth cohort of patients who had daily genital swabs performed in 23 locations (*Table 1*, *Box 4*). This study revealed that HSV-2 is present across the entire genital tract during many episodes, although viral loads are highly variable over space and time (*Figure 2A*, *Figure 2—figure supplement 1*). We next demonstrated that viral re-expansion more commonly follows a period of decay during episodes with higher peak viral loads (*Table 1*, *Box 5*, and *Figure 2B*): if each peak during a prolonged shedding episodes (*Figure 1E*) represents viral production from a single focus of infection, then this implies that multiple spatially dispersed foci of replication (*Figure 2A*) may occur due to seeding from areas with high levels of HSV-2 replication. Concurrent viral expansion and clearance in spatially distinct microregions could account for saw-tooth episodes (*Figure 1E*), and viral migration to new regions might prolong episodes.

Further evidence for this phenomenon is that herpetic lesions consist of multiple vesicles and ulcers (*Corey et al., 1983*), which occur in clusters and develop sequentially (*Figure 2C*, *Figure 2—figure supplement 2*), suggesting that virus from a single ulcer may lead to ancillary ulcer formation. In addition, we have previously demonstrated that host T-cell responses in the area of genital lesions are highly localized to foci of viral replication (*Zhu et al., 2007*, *2009*). To better classify T-cell heterogeneity over short distances, we performed biopsies in two patients from Cohort B immediately after lesion healing,

**Table 1.** Five cohorts of HSV-2 genital tract shedding

| Cohort | Subjects | Total swabs | Swabbing frequency | Total episodes | Swabbing duration | Anatomic swabbing region | Purpose |
|---|---|---|---|---|---|---|---|
| A | 3 | 96 | Every 5 min | 3 | 4 hr when lesion present | Total genital tract | Swab-to-swab sampling/assay variability |
| B | 5 | 200 | 10 times/day (every 2 hr during the days and 4 hr overnight) | 5 | 4–5 days when lesion present | Total genital tract | Episode expansion, clearance and re-expansion kinetics |
| C | 25 | 4706 | 4 times/day | 109 | 30–60 days without or with a lesion | Total genital tract | Accurate estimates for expansion/decay slopes for clinical and subclinical episodes |
| D | 2 | 216 | Daily | 4 | 30 days without or with a lesion | 23 separate regions | Spatial dispersion of HSV |
| E | 531 | 14,685 | Daily | 1020 | >30 days with or without a lesion | Total genital tract | Model fit |

**Box 1.** Cohort A

Cohort A included three subjects who were seen within 12 hr of HSV-2 recurrence, and who collected genital swabs every 5 min for 4 hr. This experiment assessed whether there were alterations in viral load related to the PCR assay or collection technique. Only small changes occurred during 5-min intervals (*Figure 1A*, *Figure 1—figure supplement 1A,B*, *Figure 1—source data 1*). The mean absolute value of viral DNA copy difference between successive swabs was 0.20, 0.28, and 0.34 $\log_{10}$ genomic copies in the three participants, and generally increased with time between swabs (*Figure 1B*, *Figure 1—figure supplement 1C*). The mean difference in HSV DNA copies between swabs correlated tightly with time between swabs in two participants (*Figure 1C*, *Figure 1—figure supplement 1D*), with virtually no difference in viral quantity between swabs separated by 5-min intervals in all three subjects (*Figure 1C*, *Figure 1—figure supplement 1D*). Two episodes had negative mean differences in viral quantity with increasing time between swabs (*Figure 1C* , *Figure 1—figure supplement 1D*), while one episode had a slightly positive mean difference (*Figure 1—figure supplement 1D*), suggesting that we captured two episodes during a decay phase and one episode near a peak of viral production. The standard deviation of the mean absolute value of viral DNA copy difference between successive swabs was 0.16, 0.22, and 0.26 in the three participants, respectively, suggesting that differences of >0.5 $\log_{10}$ HSV-2 DNA copies that may occur over hours usually reflect true changes in viral load rather than noise in the data attributable to clinical collection or PCR technique.

**Box 2.** Cohort B

To explore patterns of viral expansion and decay of an individual episode over time, we expanded the duration of evaluation and time interval between sampling. For Cohort B, we enrolled five HSV-2 positive participants with genital lesions, and sampled them 10 times daily (every 2 hr during the day and 4 hr overnight) for 4–5 days from lesion onset. A key feature of all episodes was a 'saw tooth' morphology of viral expansion and contraction (*Figure 1D*, *Figure 1—figure supplement 2*, *Figure 1—source data 1*), with 0.8–1.8 daily peaks (defined as an expansion of 0.5 $\log_{10}$ HSV DNA followed by a decline of 0.5 $\log_{10}$ HSV DNA). The mean absolute value of viral DNA copy change between successive swabs separated by 2 hr was 0.73, 0.72, 0.50, 0.51, and 0.34 $\log_{10}$ HSV DNA copies in the five participants, respectively, suggesting significant variability in viral levels. In addition, each episode was notable for high initial viral peak (6–8 $\log_{10}$ HSV DNA) followed by decay within 24 hr of initiation, highlighting extremely early rapid viral expansion, followed by rapid control.

**Box 3.** Cohort C

To expand data available for analysis of early viral expansion and late viral clearance, we performed new analyses from a published cohort of 25 subjects who were sampled every 6 hr for 30–60 days (Cohort C) (*Mark et al., 2008*). Among 109 episodes, 75% of which were subclinical, median regression slopes from initiation to peak and peak to termination were 20.3 and −8.7 $\log_{10}$/day, respectively. Median first and last copy numbers were 3.5 and 3.3 $\log_{10}$ HSV DNA copies, respectively: these measures occurred a median of 3 hr after episode initiation or before termination, suggesting that rapid changes (~10-fold/hr) in genomic copy numbers occurred both early and late within an episode (*Mark et al., 2008*). While a majority of episodes in Cohort C were <12 hr, even among the 55 episodes of >24 hr, transition from initial expansion to clearance phase occurred at a median of 12 hr, implying early immunologic pressure against the virus. Rapid expansion and contraction phases continued throughout more persistent shedding episodes. *Figure 1E* and *Figure 1—figure supplement 3* (*Figure 1—source data 1*) illustrate episodes of 17-day and 8-day duration from two participants with multiple sharp peaks from steep expansion and decay phases.

and enumerated T-cell counts at the edge of an ulcer and 1 cm away. CD8+ and CD4+ T-cell densities in the genital skin were maximal at ulcer edge (*Figure 2D*, *Figure 2—figure supplement 3A*), and considerably lower in regions 1 cm away (*Figure 2E*, *Figure 2—figure supplement 3B*). Even within a 1-mm biopsy, CD8+ T cells formed cluster-like aggregates (*Figure 2D*, *Figure 2—figure supplement 3A*), suggesting that dynamical interactions between acquired immune cells and infected cells occur within tightly spaced microenvironments on a micrometer scale. We infer that individual ulcers, both microscopic and visible to the naked eye, may occur across a wide region of the genital tract during large outbreaks. While each ulcer and microulcer represent a site of intense interface between the virus and T cells, genital skin and mucosa between ulcers may be relatively quiescent.

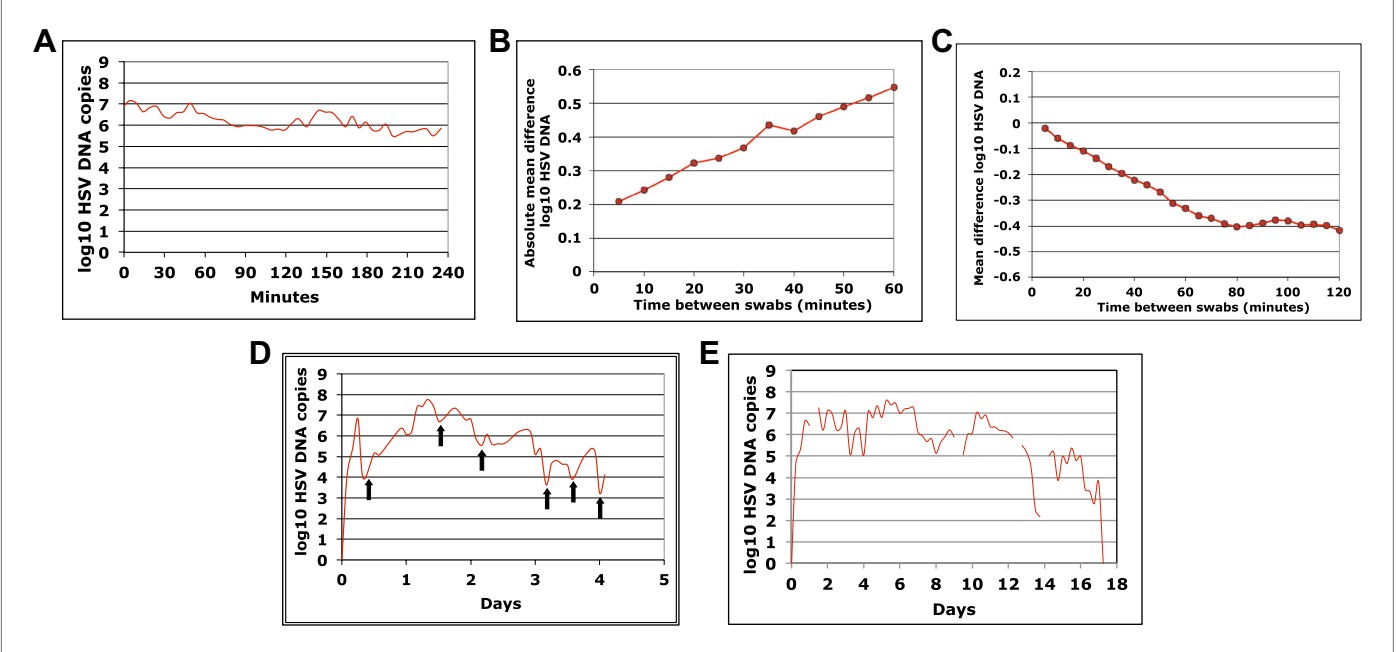

**Figure 1**. HSV-2 levels in the genital tract are stable over minutes, expand and decay markedly over hours, and fluctuate rapidly and unpredictably over days. (**A**) Shedding quantity in a participant, who performed genital swabs every 5 min over 4 hr during a lesion, reveals low swab-to-swab variation in viral quantity. Using data from panel (**A and B**), absolute mean difference ($R^2 = 0.99$), and (**C**) mean difference ($R^2 = 0.87$), in HSV DNA copies between swabs, are a function of time between swabs. (**D**) Shedding quantity in a participant, who performed 10 genital swabs per day during a lesion over 4 days (swabs every 2–4 hr), shows a characteristic saw-tooth pattern; arrows denote rapid viral re-expansion; the participant had a negative swab performed before episode onset. (**E**) Shedding quantity in a participant, who performed four genital swabs per day over 17 days demonstrates that rapid and frequent viral re-expansion allows for shedding prolongation; four missing data points are left blank.

The following figure supplements are available for figure 1:

**Source data 1.** Source data for Figure 1, Figure 1—figure supplement 1, Figure 1—figure supplement 2 and Figure 1—figure supplement 3.

**Figure supplement 1**. Dynamics of HSV-2 shedding over 5-min time intervals.

**Figure supplement 2**. Dynamics of HSV-2 shedding with every 2-4 hr sampling.

**Figure supplement 3**. Dynamics of HSV-2 shedding with every 6-hr sampling over 8 days.

---

**Box 4.** Cohort D

We investigated the spatial-temporal dynamics of reactivation in two patients in which we divided the genital tract into 23 separate regions and swabbed each individual for 30 consecutive days (Cohort D). During brief asymptomatic episodes, low HSV-2 quantities were confined to a few regions, in which viral load and location fluctuated over time (*Figure 2A*, *Figure 2—figure supplement 1*, *Figure 2—source data 1*). During longer more severe episodes, HSV was detected throughout the genital tract (*Figure 2A*), although viral density often varied by >4 logs between adjacent regions.

## Fitting dataset for mathematical models

To generate hypotheses that might explain these observations, we sought to design a mathematical model that precisely recreated the shedding patterns observed in our human shedding studies. Rather than fit models to complex individual episodes with erratic expansion and decay phases (*Figure 1D,E*), we tested competing models for their ability to reproduce key kinetic patterns from a large more generalizable patient base with genital herpes (*Table 1*, *Box 5*).

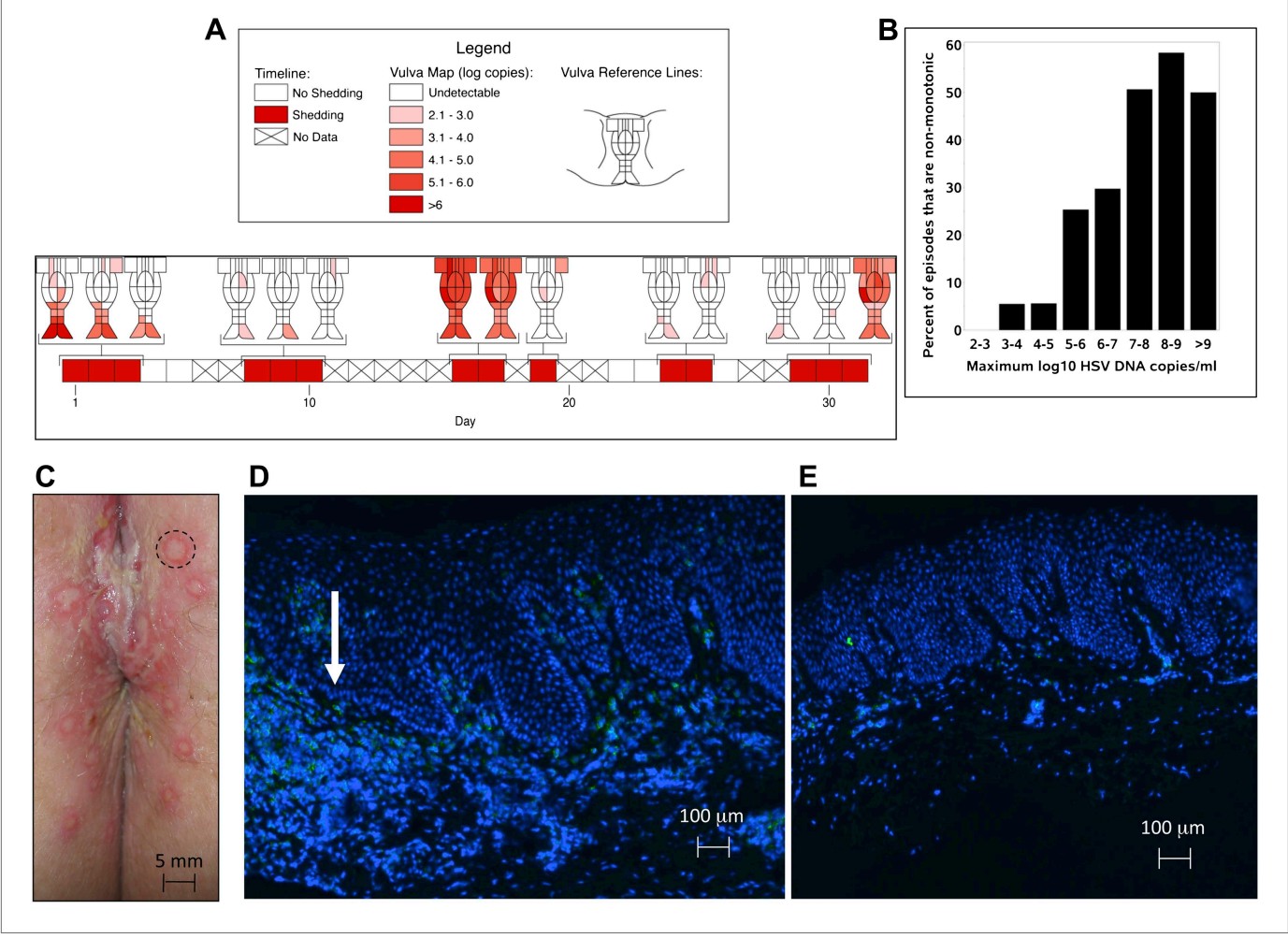

**Figure 2**. HSV-2 replicates and is contained in widely dispersed microenvironments across the genital tract. (**A**) HSV shedding quantity in a participant, who underwent daily swabs in 23 regions across the genital tract for 30 days; days without sampling are marked with an X; stars denote days with a lesion; virus is widely dispersed and several prolonged episodes with heterogeneous viral loads across the genital tract are noted. (**B**) Increasing probability of episode re-expansion (nonmonotonic episodes) as a function of peak episode copy number among 1020 episodes from 531 study subjects; individual peaks during episodes may represent virus from a single ulcer that can seed other regions. (**C**) A genital lesion consists of numerous round ulcers (black dotted circle) clustered in space; contemporaneous presence of multiple ulcers may indicate concurrent viral expansion in decay in multiple regions. (**D**) and (**E**) Immunofluorescent staining of biopsies performed (**D**) at the edge, and (**E**) 1 cm away from an ulcer 3 days post-healing; CD8+ T cells (green) at the dermal–epidermal junction (arrow) are highly localized to ulcer edge (287/mm²) and are fourfold less dense 1 cm away (72/mm²).

The following figure supplements are available for figure 2:

**Source data 1.** Source data for Figure 2 and Figure 2—figure supplement 1.

**Figure supplement 1**. Spatial features of HSV genital tract shedding.

**Figure supplement 2**. Spatial features of HSV-2 lesions.

**Figure supplement 3**. Spatial features of CD8+ T-cell response in genital skin.

## Spatial model of HSV-2 dynamics

We attempted to develop a model that could precisely capture episode heterogeneity within and between persons by recreating frequency histograms from Cohort E, and predicting key observations from Cohorts A–D, including rapid viral expansion and decay, multiple peaks per episode, and wide

**Box 5.** Cohort E

To define episode heterogeneity, we used the largest available database of HSV-2 shedding (Cohort E), which contained HSV PCR data taken from 531 HSV-2 seropositive persons who sampled the entire genital tract once daily for >30 days. This dataset included 14,685 swabs and 1020 episodes of symptomatic and asymptomatic shedding (*Schiffer et al., 2011*). Each episode was classified according to six features: duration, first, peak and last positive HSV genomic copy number, and initiation to peak and peak to termination slopes. The heterogeneity of each feature was captured using frequency histograms. Shedding episodes had highly variable duration (median days: 3; IQR: 1–8) and peak HSV DNA copy number (median $\log_{10}$ copy number: 4.8; range: 2–9.2). Episode rate was 15.3/year, and per swab shedding frequency was 18% (~3% of swabs contained $10^2$–$10^3$, $10^3$–$10^4$, $10^4$–$10^5$, $10^5$–$10^6$, $10^6$–$10^7$, and >$10^7$ HSV DNA copies, respectively). In Cohort E, 19% of episodes had re-expansion, which we previously defined as 0.5 log decay followed by 0.5 log re-expansion (*Schiffer et al., 2011a, 2011b*).

spatial distribution of HSV DNA. After several iterations of model development failed to reproduce key features of our empirical dataset ('Methods'), we designed a spatial model that accounted for observed multiple concurrent foci of viral replication and immune response (*Figure 2*). We hypothesized that secondary ulcer formation in separate genital tract regions might occur via two mechanisms: release of HSV-2 from neuron endings may simultaneously occur in spatially distinct genital regions, or dense aggregates of cell-free particles within an ulcer may be locally infectious. Evidence for the latter hypothesis is tight correlation between episode peak copy number and probability of re-expansion in (*Figure 2B*). In addition, ulcers tend to be closely arrayed in space rather than widely dispersed across vulnerable regions in the genital tract. 'Kissing lesions', in which two ulcers form at the site of skin-to-skin contact, are a well-documented phenomenon (*Figure 2C*). On the other hand, ulcers do not isolate within genital tract dermatomes as occurs with zoster reactivation. These observations suggest that local seeding from epithelial lesions may be responsible for many secondary ulcers.

Our spatial model included 300 regions of 6.5 mm diameter linked in a spatial arrangement of adjacent hexagons (*Figure 3A*, *Figure 3—figure supplement 1*). The diameter was chosen as an absolute maximal value for ulcer diameter. We populated each region with differential equations (*Figure 3B*, 'Methods') describing infected epithelial cells (I), which produced cell-associated HSV-2 ($V_i$) at a rate $p$. Cell-associated HSV-2 was infectious to susceptible epithelial cells (S) according to an infectivity parameter $\beta_I$. Infected cells died either due to direct HSV-2 killing at a rate $a$, or due to CD8+ T-cell killing at a rate ($f \times E$). Cytolytic CD8+ T cell (E) expanded at a maximal rate $\theta$. CD8+ expansion rate increased according to number of infected cells, and was half-maximal ($\theta/2$) at a threshold value of infected cells, $r$. Cell-associated HSV-2 converted to cell-free HSV-2 ($V_e$) following cell lysis. Cell-free viruses and CD8+ T cells decayed at fixed rates ($c$ and $\delta$) within each region. We assumed that viruses ($V_{neu}$) were randomly released into 300 regions by neurons at a rate $\phi$, predicted by a previous model (*Schiffer et al., 2009*), and that these viruses could initiate an ulcer in each reason by infecting an epithelial cell.

Adjacent regions in the model were linked virally. Cell-associated HSV ($V_i$) drove spread within an ulcer in a single region, while cell-free HSV ($V_e$) could initiate new ulcers at infectivity $\beta_e$, but only in six contiguous regions surrounding a productive ulcer (*Figure 3A*, *Figure 3—figure supplement 1*). Based on our observation in cell culture that in a single cell infected by a single virus, viral replication does not occur until approximately 12–16 hr, a fixed time delay parameter ($\epsilon$) was included for ulcer formation.

The physical distance between regions was not explicitly considered because the 300 regions were not intended to capture the complex three-dimensional topography of genital skin. Rather, the distance between regions was captured in immunologic terms. Based on the gradient of CD8+ T-cell density as distance increases from an ulcer edge (*Figure 2D,E*), we assumed that contiguous regions might be immunologically codependent, by including a new fitting parameter ($\rho$) to estimate the extent that CD8+ T-cell density in contiguous regions affected CD8+ T-cell density within a new ulcer region ('Methods'). Contiguous regions in the model were therefore assumed to be far enough away for new ulcers to initiate but potentially close enough to be effected by neighboring immune responses.

## Model fitting

We solved our model by fitting to the data and assuming either 5 or 10 above parameter values as unknown ('Methods'). In both cases, model output closely reproduced the data within Cohort E, including quantitative shedding frequency (*Figure 4A*), as well as episode rate (*Figure 4B*), median initiation to peak

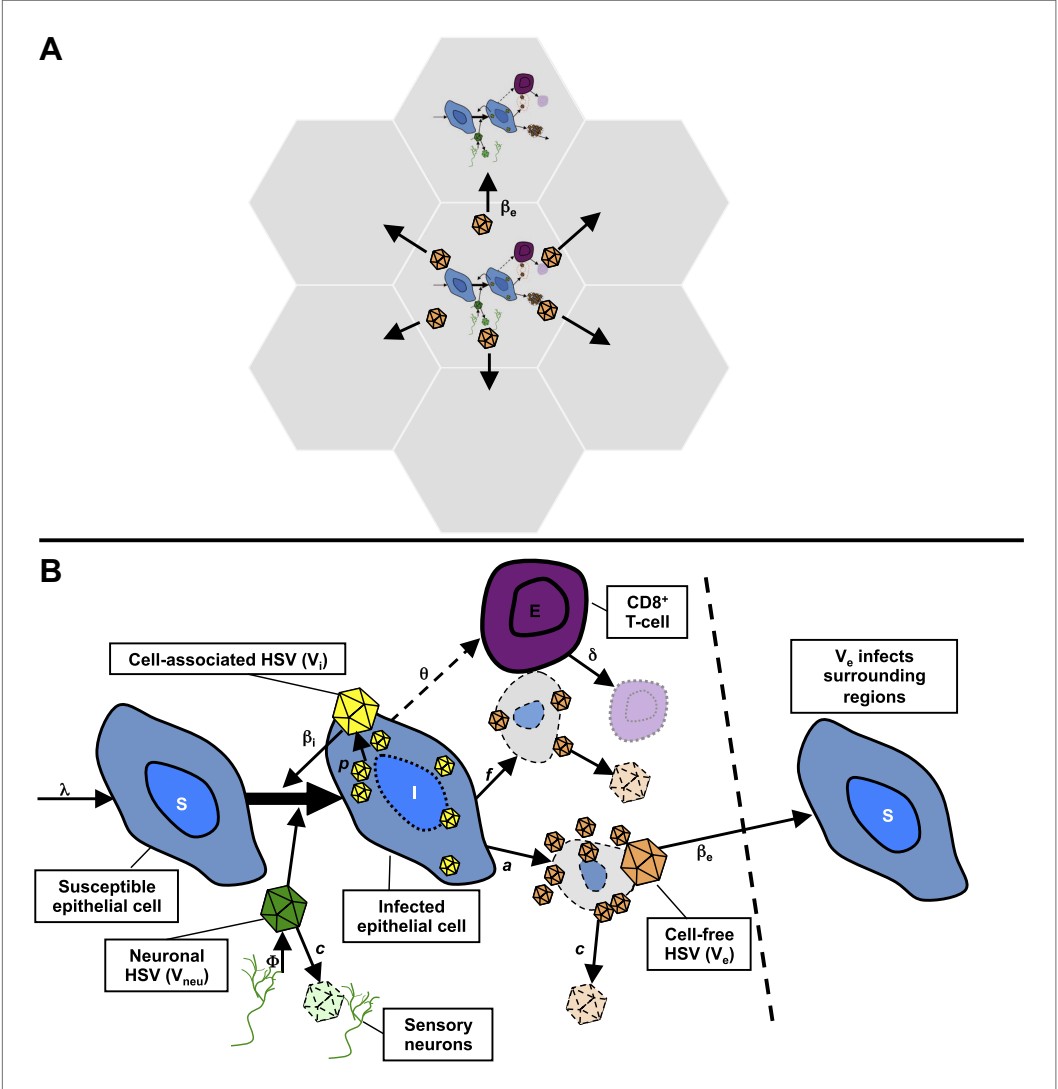

**Figure 3**. Mathematical model. (**A**) Microregions are linked virally because cell-free HSV-2 can seed surrounding regions, and immunologically based on overlapping CD8+ T-cell densities between regions (not shown). (**B**) Schematic for HSV-2 infection within a single genital tract microenvironment. Equations capture seeding of epithelial cells by neuronal HSV-2, replication of HSV-2 within epithelial cells, viral spread to other epithelial cells, cytolytic CD8+ T-cell response to infected cells, transition of cell-associated HSV-2 to cell-free HSV-2 following lysis of infected cells, and elimination of free virus and infected cells.

The following figure supplements are available for figure 3:

**Figure supplement 1**. Spatial mathematical model.

and peak to termination slopes (*Figure 4C*), durations (*Figure 4D*), and first (*Figure 4E*), last (*Figure 4F*), and peak HSV DNA copy numbers (*Figure 4G*, *Figure 4—source data 1*). We also performed a sensitivity analysis using 500 episode (~30 years) simulations in which single parameter values were adjusted to arrive at narrow ranges for parameter values that reproduced our data (*Table 2*). These parameter values were generally within an order of magnitude of previous parameter estimates (*Schiffer et al., 2009*).

## Frequent heterogeneous episodes with extremely rapid expansion, rapid decay, and multiple peaks

The model was next evaluated for its ability to predict other key features of genital shedding identified in Cohorts A–D. To explore the dynamics of simulated episodes with higher granularity, we varied

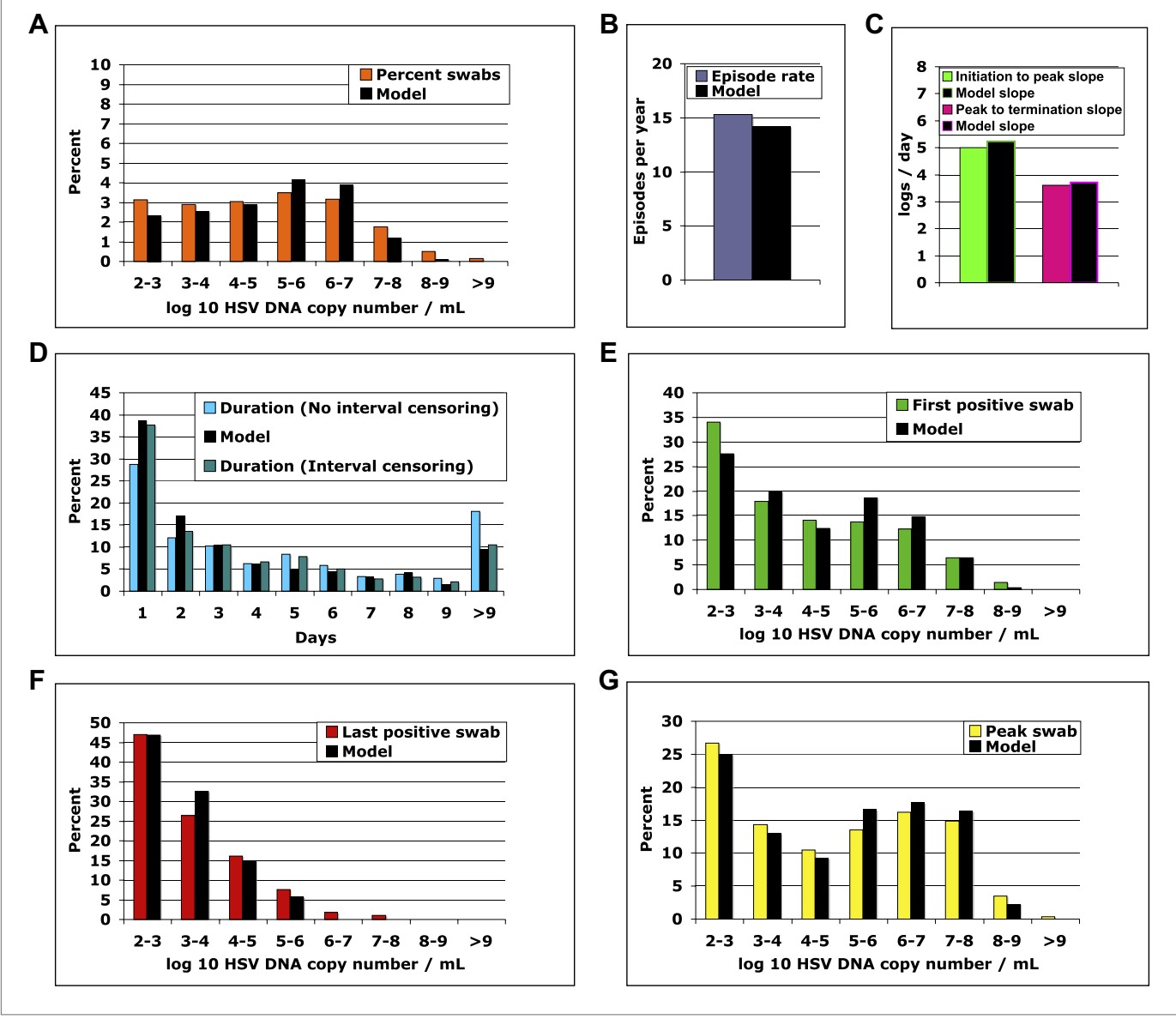

**Figure 4**. The spatial model reproduces all shedding episode characteristics. Colored bars represent results from (**A**) 14,685 genital swabs and (**B**–**G**) 1020 shedding episodes from 531 study participants. The model simulation, represented with black bars in each panel, continued until 1020 episodes were generated; model sampling occurred every 24 hr as in the clinical protocol. Model output reproduced (**A**) quantitative shedding frequency as well as (**B**) rate, (**C**) median initiation to peak and peak to termination slopes, (**D**) Duration, (**E**) first HSV DNA copy number, (**F**) last HSV DNA copy number, and (**G**) peak HSV DNA copy number of episodes.

The following figure supplements are available for figure 4:

**Source data 1.** Source data for Figure 4.

**Figure supplement 1**. Continuous sampling of spatial model output reveals more accurate measures of episode characteristics.

sampling frequency of model output to include continuous sampling. The model predicted the empirically derived finding that daily sampling substantially underestimates expansion (*Figure 4— figure supplement 1A*) and clearance slopes (*Figure 4—figure supplement 1B*). Among 842 simulated episodes with sampling every 0.001 days, median initiation to peak expansion rate was 25.5 $\log_{10}$ HSV DNA copies per day (versus 20.3 $\log_{10}$ copies per day with 6-hr sampling in Cohort C),

**Table 2.** Parameter ranges that result in accurate reproduction of model outcomes

| Parameter | Units | Symbol | Best fit value | Good fit Lower limit | Good fit Upper limit | Average fit Lower limit | Average fit Upper limit |
|---|---|---|---|---|---|---|---|
| Cell-associated HSV infectivity | DNA copy days/cell (viruses needed per day to infect one adjacent cell) | $\beta_i$ | $5.4e^{-8}$ (111) | $4.86e^{-8}$ (123) | $7.83e^{-8}$ (76) | $3.78e^{-8}$ (158) | $1.32e^{-7}$ (45) |
| Cell-free HSV infectivity | DNA copy days/cell (viruses needed per day to initiate one new ulcer) | $\beta_e$ | $2.65e^{-11}$ ($2.26e^5$) | $1.73e^{-11}$ ($3.46e^5$) | $2.78e^{-11}$ ($2.15e^5$) | $3.98e^{-12}$ ($1.50e^6$) | $5.04e^{-11}$ ($1.19e^5$) |
| Epidermal HSV replication rate | HSV DNA copies per cell per day | $p$ | $1.03e^5$ | $7.21e^4$ | $1.7e^5$ | $5.15e^4$ | $1.96e^5$ |
| Neuronal release rate | HSV DNA copies per day per genital tract | $\phi$ | 82 | 45 | 90 | 41 | 123 |
| Free-viral decay rate | Per day (half-life, hours) | $c$ | 8.8 (1.9) | 7.0 (2.4) | 9.7 (1.7) | 6.2 (2.7) | 12.3 (1.4) |
| Maximal CD8+ T-cell expansion rate | Per day | $\theta$ | 2.84 | 1.85 | 3.27 | 1.85 | 5.25 |
| CD8+ T-cell decay rate | Per day (half-life, days) | $\delta$ | $1.47e^{-3}$ (471) | $1.12e^{-3}$ (619) | $1.69e^{-3}$ (409) | $6.64e^{-4}$ (1040) | $2.21e^{-3}$ (314) |
| CD8+ T-cell local recognition | Infected cells at which $\theta$ is half maximal | $r$ | 42 | 30 | 47 | 4 | 74 |
| CD8+ regional codependence | 0 = no codependence, 1 = full codependence | $\rho$ | 0.69 | 0.59 | 0.86 | 0.38 | 0.86 |
| Viral production lag | Days | $\varepsilon$ | 0.96 | 0.53 | 1.1 | 0.34 | 1.1 |

implying that during the early expansion phase, HSV DNA levels increased 10-fold every 57 min and doubled every 17 min. There was also rapid late clearance ($-7.4$ $\log_{10}$ copies per day vs $-8.7$ $\log_{10}$ copies per day in Cohort C).

We simulated the model until 100 episodes with >10,000 total infected cells within one ulcer were generated. Number of infected cells and viral load peaked at a median of 13.5 hr, predicting the finding from Cohort C that transition from expansion to clearance occurs ~12 hr after episode commencement. To explain episodes of >4 days, the model predicted multiple peaks within single episodes. In addition, prolonged simulated episodes had wide concurrent dispersal of virus (*Video 1*).

While daily sampling accurately measured quantitative shedding frequency with continuous sampling (*Figure 4—figure supplement 1C*), the model predicted that more frequent sampling increased episode detection by ~50% (*Figure 4—figure supplement 1D*) and proportion of episodes with low duration (*Figure 4—figure supplement 1E*) and low peak copy number (*Figure 4—figure supplement 1F*). The model also predicted the finding from Cohort C that ~75% of HSV-2 episodes are asymptomatic, brief (<48 hr), and associated with a low peak copy number

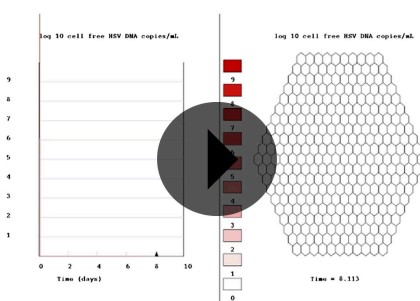

**Video 1**. Spatiotemporal demonstration of a 10-day episode. The left panel represents total cell-free HSV DNA copies per milliliter present over time. The right panel represents spatial spread of virus during the episode, each hexagon contains one region of shedding and virus spreads to contiguous regions. Amount of virus within a single region is displayed according to a heat map adjacent to the spatial map.

(*Figure 4—figure supplement 1F*) (*Mark et al., 2008*). There were higher absolute numbers of episodes with high copy number (>$10^7$ HSV DNA copies) because daily sampling rarely captured true episode peak (*Figure 4—figure supplement 1G*).

## Efficient cell-to-cell HSV-2 spread within a single epidermal ulcer

Our estimates of parameter values suggest that individual cell-associated viruses are rapidly produced by epidermal cells and are highly infectious to surrounding cells. Epidermal cells produced 72,000–170,000 HSV DNA copies per day; one of 76–123 cell-associated DNA particles infected a neighboring cell per day at ulcer onset (*Table 2*). When we simulated the model until 100 episodes with >10,000 total infected cells within a single ulcer were generated, median times to 100, 1000, and 10,000 total infected cells were 5.6, 8.7, and 12.9 hr, respectively. Median time interval from becoming infected to infecting a new cell during the expansion phase of the 100 simulated episodes was 6.8 hr; median time from becoming infected to infecting a first surrounding cell was 59 min. This finding was surprising because in culture, HSV replication occurs within newly infected cells 4–6 hr post-infection (*Roizman, 2007*). Our data suggest that in vivo an HSV-infected keratinocyte might become locally infectious by spreading the virus even before replication initiates within this cell, a phenomenon recently described for vaccinia (*Doceul et al., 2010*).

## Episode prolongation due to secondary ulcer formation

The model results suggested that secondary ulcer formation by cell-free particles is crucial for episode prolongation. In 500 simulated episodes, duration was a function of total ulcers (*Figure 5A*). We analyzed a simulated prolonged episode (10 days) in detail to assess kinetics within single ulcers: free viral production from the initial viral ulcer lasted 3 days (*Figure 5B*); infected cells were eliminated within 24 hr (*Figure 5C,D*); peak viral load and number of infected cells were 8.1 $\log_{10}$ HSV DNA copies and 4.1 $\log_{10}$ cells, respectively, although total amount of viral DNA and infected cells produced within the ulcer were considerably higher (8.5 $\log_{10}$ HSV DNA copies and 4.9 $\log_{10}$ cells, respectively), reflecting rapid viral and infected cell turnover. Virions generated during the primary ulcer promoted formation of additional ulcers (*Figure 5E* and *Videos 1, 2*). Before episode termination, 24 total ulcers were produced. The median duration and peak viral production of the 24 ulcers were 2.5 days (range: 0.6–4.6 days) and 5.2 $\log_{10}$ HSV DNA copies (range: 2.0–8.0 $\log_{10}$), respectively. We calculated diameter of each ulcer during simulations based on the number of missing cells due to death from infection, and defined a lesion as any episode with a >1 mm ulcer (*Schiffer et al., 2009*). Consistent with clinical observations in Cohort D, simulated ulcer size did not exceed 5 mm (*Corey et al., 1983*), and many failed to reach the clinical threshold of 1 mm in diameter (*Video 3*).

Parameter estimates suggest that cell-free HSV was ~2000-fold less infectious than cell-associated HSV (*Table 2*), possibly because HSV in the aqueous environment outside of the cell must contend with genital secretion flow kinetics, mucous, low pH, and neutralizing or ADCC-like antibodies, while cell-associated HSV avoids these hazards by congregating within cellular tight junctions (*Collins and Johnson, 2003*). In nonmucosal regions, a layer of dead cells that do not support viral replication protects underlying nucleated keratinocytes in the epidermis against infection, further decreasing the probability that cell-free HSV will contact a viral entry receptor.

Due to slower elimination of cell-free particles, the model predicted that the number of ulcers containing cell-free particles often exceeded that of ulcers populated by HSV-infected cells, thus prolonging the opportunity for secondary seeding (*Video 3*). To address this question, we conducted a 365-day simulation with infectious and noninfectious cell-free HSV DNA (*Videos 4 and 5*) and found that noninfectious cell-free HSV, as may occur in the presence of comprehensive neutralizing antibody protection, resulted in a decrease in episode duration and shedding frequency from 18.2% to 10.8%.

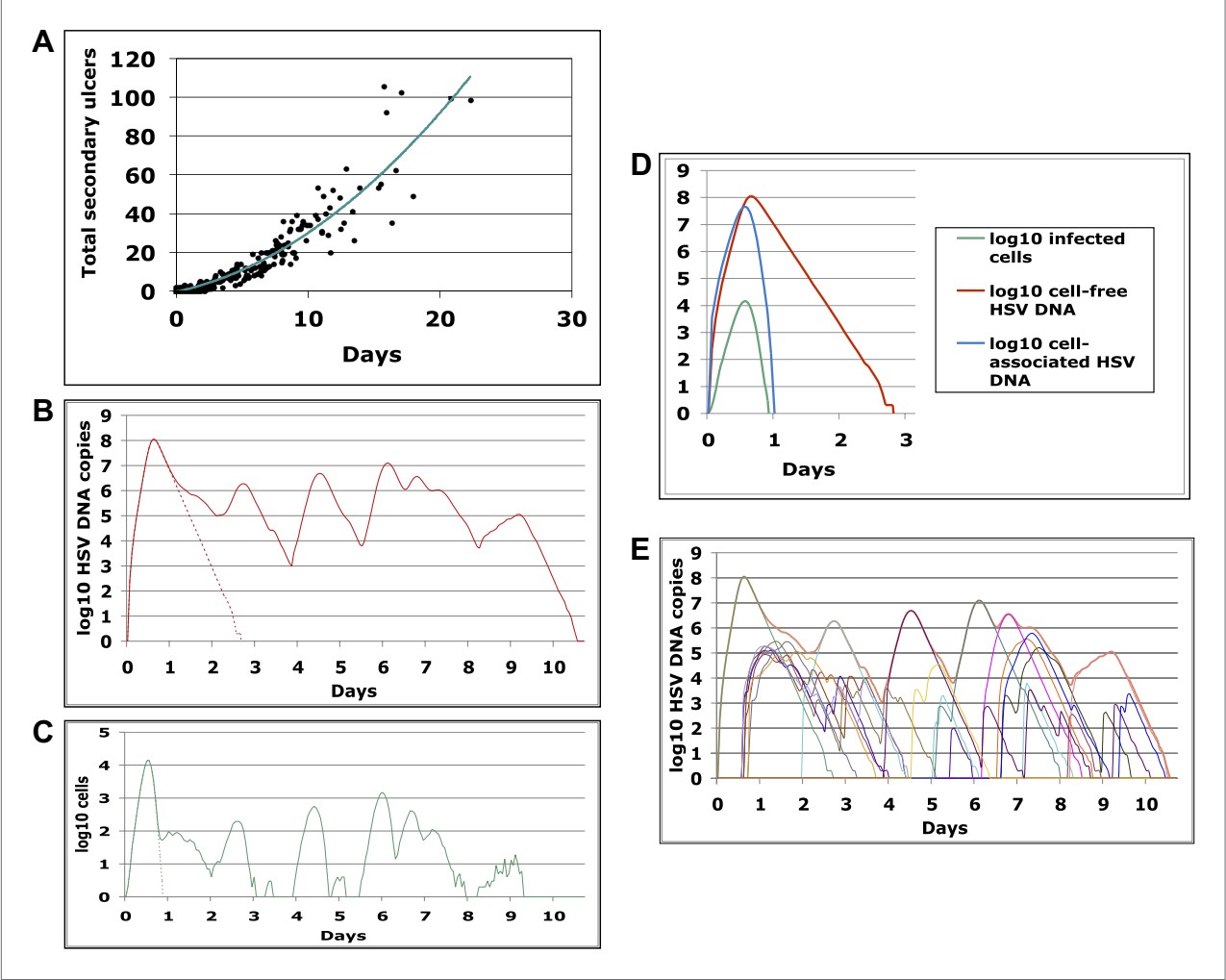

**Figure 5**. Containment of infected cells within a single ulcer is extremely rapid, although secondary ulcers explain prolonged episodes. (**A**) Episode duration was a function of the number of ulcers before episode termination during 500 simulated episodes. (**B**)–(**E**) A 10-day simulated episode consisting of 24 ulcers: (**B**) Total cell-free virus (red) over time reflects the saw-tooth pattern of prolonged episodes; virus produced from the initial ulcer (red dotted line) was eliminated within 3 days. (**C**) Infected cells were eliminated from the initial viral ulcer (green dotted line) within 1 day and there were four periods during the episode when no infected cells were present. (**D**) Cell-free virus (red), cell-associated virus (blue), and infected cells (green) were eliminated from the primary ulcer with different kinetics; infected cells peaked at 13 hr and were extinguished in <24 hr (**E**) Secondary ulcers prolonged episodes; each thin line represents HSV-2 production from a specific region.

## Localized CD8+ T-cell density as a key determinant of episode severity

In spatial simulations, despite rapid initial accumulation, infected cells were eliminated rapidly, highlighting the intense local immune response (*Figure 5C,D*). In the 100 simulated episodes described above, the median time to death of 10,000, 1000, or 100 infected cells was 13.7, 9.7, or 6.9 hr, respectively. We previously described an inverse relationship between HSV-2 episode severity and CD8+ T-cell density at the site of reactivation (*Schiffer et al., 2010*). Our models predict that CD8+ T-cell density is an inverse correlate of the reproductive number (the number of cells infected by the initially infected cell) in a region. Once T-cell density reaches a certain threshold, R falls below 1 and HSV-containing cells are cleared before infection of 10 cells can occur (*Schiffer et al., 2010*).

We performed a global sensitivity analysis of 500 simulated episodes ('Methods') to evaluate drivers of single episode severity. CD8+ T-cell density within the initially infected region was the most predictive parameter of episode peak viral load and duration. Increased viral replication rate in epidermal cells and decreased CD8+ T-cell expansion rate also correlated with higher peak viral

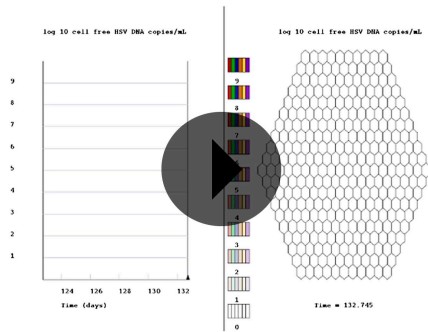

**Video 2.** Spatiotemporal demonstration of a 14-day episode according to viral production within each single region. The left panel represents cell-free HSV DNA measured over time with each region's production demonstrated with a different color. The right panel represents spatial spread of virus during the episode; colors in the right panel correspond to those in the left panel. Amount of virus within a region is displayed according to a heat map adjacent to the spatial map.

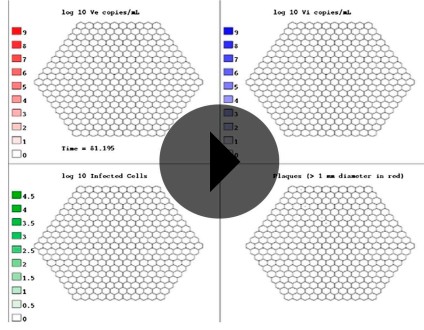

**Video 3.** Spatiotemporal demonstration of infected cell and viral spread during a 6-day episode. The upper left panel represents spatial spread of cell-free virus during the episode; the upper right panel represents spatial spread of cell-associated virus during the episode; the bottom left panel represents spatial spread of infected cells during the episode; the bottom right panel represents ulcer formation during the episode, ulcers turn from black to red when diameter exceeds 1 mm; quantities are displayed according to a heat map adjacent to each spatial map. There is more rapid decay of infected cells and cell-associated particles than of cell-free particles within each region. Visible ulcers persist after viral production terminates within a region.

production, while increased viral infectivity and replication rate correlated with longer episode duration (**Table 3**).

These findings suggest that local T-cell density at the site of neuronal HSV-2 release into genital skin determines short-term viral trajectories. When high CD8+ T-cell density regions enclosed the initial infected region, spread was limited (**Video 6**), whereas more prolonged episodes followed a serpiginous route along low CD8+ T-cell density pathways before encountering a high-density dead end (**Video 7**). Episodes of >10 days occurred when large genital tract areas were at low CD8+ T-cell density (R > 1) (**Video 8**). HSV replication sometimes terminated at edge regions, which is biologically realistic because herpetic lesions rarely form in skin surrounding the genital tract.

After prolonged episodes, portions of the genital tract were effectively immunized against high-copy episodes for a short period, while other regions remained susceptible to substantial spread of virus. Over years, CD8+ T-cell and reproductive number spatial patterns cycled intermittently (**Video 9**), suggesting that HSV causes a chronic dynamic inflammatory state in the genital tract. These predictions are consistent with empirical studies of biopsied tissue of HSV-2–infected patients (**Cunningham et al., 1985**; **Zhu et al., 2007**, **2009**), and may explain relatively stable genital shedding rates seen within infected persons over decades.

## Widely dispersed spatial drip of HSV-2 from neurons into genital tissue

To better define viral reactivation and latency patterns that best explain our shedding datasets, we altered the frequency and spatial dispersion of neuronal HSV release in the model; genital shedding trends were unchanged if HSV was released continuously or in daily, every 3-day, or weekly pulses, provided that the average HSV DNA release rate remained ~82/day, and that release occurred randomly throughout the genital tract (**Figure 6**). Simulations in which release occurred continuously at 82 HSV DNA particles per day into 1, 5, or 20 randomly selected regions resulted in a poor fit, due to a higher percentage of rapidly contained episodes and lower shedding rate (**Figure 6—figure supplement 1**). HSV-2 release into 50 geographically clustered regions within a confined area led to lower episode rate, less shedding, and poorer overall fit (**Figure 6—figure supplement 2**). These data suggest that neuronal HSV release occurs at a frequent slow trickle across wide regions of the genital tract.

## Determinants of intersubject variability in shedding and lesion rates

We next tested the model's ability to predict substantial variability in shedding, lesion, and episode rates in studies consisting of 30- to 60-day sampling periods (**Wald et al., 1995**, **1997**, **2000**; **Crespi et al., 2007**; **Mark et al., 2008**). Even when parameter values were

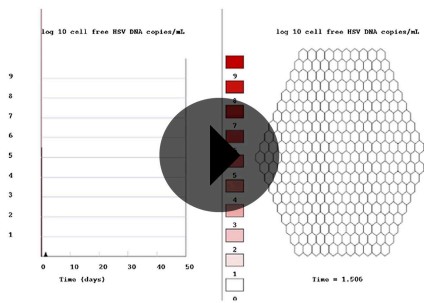

**Video 4**. Spatiotemporal demonstration of 365 days of simulated shedding. The left panel represents total cell-free HSV DNA copies per milliliter present over time. The right panel represents spatial spread of virus during the episode; each hexagon contains one region of shedding and virus spreads to contiguous regions. Amount of virus within a single region is displayed according to a heat map adjacent to the spatial map. The simulation is notable for episodes of variable duration and peak HSV DNA copy number. Prolonged episodes at days 8, 38, 136, 158, and 288, display several re-expansion phases.

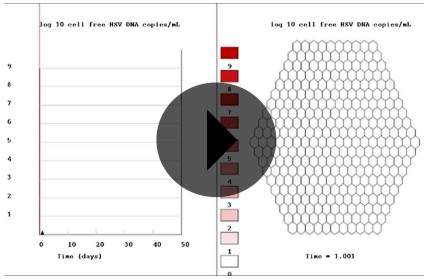

**Video 5**. Spatiotemporal demonstration of 365 days of simulated shedding with noninfectious cell-free particles. The left panel represents total cell-free HSV DNA copies per milliliter present over time. The right panel represents spatial spread of virus during the episode; each hexagon contains one region of shedding. Amount of virus within a single region is displayed according to a heat map adjacent to the spatial map. The simulation is notable for lack of prolonged episodes and lack of episode re-expansion.

not varied, the stochastic model produced realistic outcomes (**Table 4**). Lesion and shedding frequency were variable with 30- and 60-day simulations, corresponding to previously published results (**Wald et al., 1997**, **2000**; **Mark et al., 2008**). Simulations of 1 and 10 years, generated less varied outcomes (**Table 4**), suggesting that the random nature of episode initiation, along with short sampling periods may account for some shedding heterogeneity seen in clinical studies.

To assess other possible hypotheses for contrasting shedding rates, we conducted another global sensitivity analysis in which 500 Monte Carlo simulations were conducted over a 10-year time frame ('Methods'). Predictors of elevated shedding rate included high neuronal release rate of HSV-2 DNA, high CD8+ T-cell replication rate, high regional co-dependence of CD8+ T-cell density, low HSV-2 DNA replication rate in skin cells, and low free-viral decay rate (**Table 3**). The finding that parameter levels which would appear to favor immune control in the mucosa, in fact correlate with higher long-term shedding rates, highlight the complex nonlinear dynamics of the spatial model; parameters that favor lower amounts of viral replication in the short term promote a higher proportion of episodes that are contained rapidly by the immune system; CD8+ T-cell expansion is less substantial during these brief episodes. Therefore, episodes are more commonly initiated at lower CD8+ T-cell densities. As a result, conditions of high CD8+ T-cell replication rate and low HSV-2 DNA replication rate in keratinocytes tend to favor both a higher proportion of very brief episodes (<1 day) and more severe episodes that spread to multiple regions. The overall impact is to increase the shedding rate.

## Discussion

We used a strategy of intense sampling over intervals ranging from every 5 min to every 24 hr, to characterize temporal and spatial dynamics of genital HSV-2 replication. While we previously identified that a majority of HSV-2 reactivations last less than a day (**Mark et al., 2008**), our current findings demonstrate that the cardinal feature of HSV-2 shedding is extraordinary episode heterogeneity. Longer and higher copy number episodes, which often manifest with genital lesions, are notable for multiple erratic peaks and wide viral dispersion. Viral loads are stable over periods of minutes but can change dramatically over hours. HSV-2 expansion and decay phases in mucosa are considerably more rapid and complex than those of HIV and hepatitis B, which have been characterized by sampling plasma (**Ciupe et al., 2007**; **Ribeiro et al., 2010**). To explain these findings, we developed a stochastic spatial model that accurately reproduces empirically derived episode rate, duration, early viral expansion, late viral decay rate, and viral re-expansion frequency, as well as serpiginous spatial distribution of viral ulcers during recurrences.

Our findings redefine the pace of viral immune interaction within a single herpetic ulcer. HSV replication within epithelial cells, and spread by cell-to-cell transfer, seem to occur extremely rapidly and efficiently. The model predicts that >10^8 viral DNA genomes can be produced in one ulcer,

**Table 3.** Predictive model parameters for key model outcomes

| | Single episode features | | Long-term shedding features | |
| --- | --- | --- | --- | --- |
| | Peak viral load | Duration | Shedding rate | Episode rate |
| CD8+ T-cell density at reactivation site | −0.56 | −0.47 | NA | NA |
| Cell-associated HSV infectivity | — | 0.12 | — | 0.13 |
| Cell-free HSV infectivity | — | — | — | — |
| Epidermal cell replicate rate | 0.13 | 0.14 | −0.25 | −0.31 |
| Neuronal release rate | — | — | 0.43 | 0.55 |
| Free-viral decay rate | — | — | −0.2 | — |
| Maximal CD8+ T-cell expansion rate | −0.09 | — | 0.37 | 0.51 |
| CD8+ T-cell decay rate | — | 0.09 | — | −0.16 |
| CD8+ T-cell local recognition | — | — | — | — |
| CD8+ regional co-dependence | — | — | 0.32 | 0.34 |
| Viral production lag | — | — | 0.24 | 0.23 |

Partial correlation coefficients are listed only for parameters that are found to improve predictive effect on outcomes using Akaike information criteria models. Episode features are from 500 single episode simulations. Long-term shedding outcomes were measured over 10-years during 500 simulations.

even though the immune response becomes dominant only ~13.5 hr after HSV-2 detection. Routes of spread between tightly packed epithelial cells might include transit through tight cellular junctions (*Collins and Johnson, 2003*; *Farnsworth and Johnson, 2006*), distant propulsion when cells violently lyse, and homogenous mixing with susceptible cells as a result of high viral titers packed inside herpetic vesicles (*Spruance et al., 1977*).

**Video 6**. Spatiotemporal demonstration of immune response to a pair of 2-day episodes. The upper left panel represents total cell-free HSV DNA copies per milliliter present over time. The upper right panel represents spatial spread of cell-free virus during the episode. The lower left panel represents CD8+ T-cell density within each region. The lower right panel indicates reproductive number within each region. Quantities are displayed according to a heat map adjacent to each spatial map. Areas with high CD8+ T-cell levels and low reproductive numbers do not support high-level viral production. The simulated episodes are short because virus does not spread from the initial plaque to adjacent regions with high CD8+ T-cell density and reproductive numbers less than one.

The model is in agreement with mouse studies, which show that tissue-resident CD8+ T-cell density at the site of viral challenge dictates lesion severity (*Gebhardt et al., 2009*; *Shin and Iwasaki, 2012*), and further suggests that local CD8+ T-cell count supersedes all other parameters of viral replication and immunologic response in determining the extent of shedding during an episode. While common wisdom is that extended viral replication occurs due to delayed arrival of acquired immune cells to the site of viral pathogenesis, our observations suggest that the immune response in microregions of replication is extraordinarily rapid. Most shedding episodes are contained within hours, when <10 cells are infected (*Schiffer et al., 2009*). Yet, even if thousands of epithelial cells get infected in an ulcer, these cells appear to be eliminated in <24 hr.

Our model suggests that cell-free virus plays an important role in episode prolongation: cell-free viruses seed adjacent regions where T-cell density may be lower, leading to the formation of secondary foci of viral replication, which allows wide spread of virus across the genital tract. Thus, the spatial dynamic between virus and host defines episode morphology, and explains variability in shedding severity in an individual over periods of hours to weeks. These results suggest that studies of neutralizing and antibody-dependent cell-mediated cytotoxicity responses in localized mucosal areas are needed, and may explain why antiviral therapies which only impact intracellular replication and do not eliminate prolonged episodes.

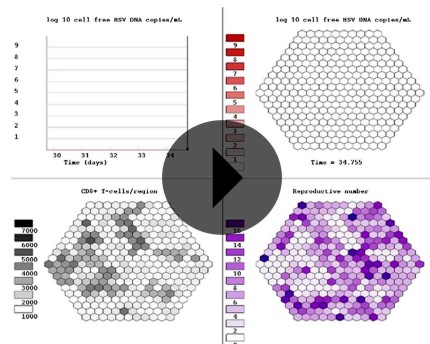

**Video 7.** Spatiotemporal demonstration of immune response to a 7-day episode. The upper left panel represents total cell-free HSV DNA copies per milliliter present over time. The upper right panel represents spatial spread of cell-free virus during the episode. The lower left panel represents CD8+ T-cell density within each region. The lower right panel indicates reproductive number within each region. Quantities are displayed according to a heat map adjacent to each spatial map. The simulated episode is medium length because virus spreads from the initial region to adjacent regions with low CD8+ T-cell density and reproductive numbers less than one, but is ultimately contained when it reaches an anatomic edge region outside of the genital tract, and regions of high CD8+ T-cell density and low reproductive number within the genital tract.

While the model successfully generates hypotheses to explain complex shedding patterns in humans, it also highlights the difficulty of predicting an individual's short-term shedding pattern, and in particular, the timing and morphology of their next episode. A heterogeneous spatial array of immune cell densities at episode onset may determine the complex trajectory of a shedding episode, but these starting conditions cannot be directly measured in humans. It is similarly not possible to know in which specific microregion an episode will initiate. When we simulated the model over 30-day intervals typically used in our clinical studies, there was heterogeneity in the shedding rate based solely on randomness of episode initiation and high variability of episode duration. As such, we did not employ well-validated methods to infer parameter values by fitting to an individual's nonlinear shedding data (*Ionides et al., 2006*).

On the other hand, the model does raise intriguing possible explanations for observed heterogeneity in long-term shedding behavior between infected persons (*Wald et al., 1997*). Surprisingly, while enhanced immune function in the form of lower viral replication rate or viral infectivity has the predictable short-term effect of limiting episode duration, the model suggests that sustained higher T-cell CD8+ replication rate and lower viral replication rate might counterintuitively increase long-term shedding rates: these parameter conditions favor a higher proportion of rapidly contained episodes, which in turn lowers the degree of CD8+ expansion. Therefore, the CD8+ density at episode initiation is on average lower, which also allows for more frequent widely dispersed episodes. Because the overall stability of parameter values over time cannot be easily measured, and because it is unknown if the model's parameters behave independently or in a colinear fashion, these sensitivity analysis predictions are relevant for hypothesis generation only. An area of prospective research is to identify which virological and immune factors account for phenotypic variability between infected persons.

For the model to precisely recapitulate episode rate, it is necessary to assume a nearly constant slow drip of HSV-2 from neurons (*Schiffer et al., 2009*). The stochastic model and spatial shedding data both suggest that there is diffuse dispersal of virus from neurons into genital epithelia, rather than focal reactivations only within certain microregions. Such a strategy allows HSV-2 to randomly seed regions where immune cell decay from previous reactivations may allow a high-titer episode to initiate. This does not imply that latency is constantly bypassed within all neurons. Rather, we favor the idea that the vast majority of HSV within ganglia is maintained in a low copy number latent state (*Wang et al., 2005*; *Verjans et al., 2007*), while a minority reactivates albeit on a frequent basis. More experimental work is needed to precisely define the true mechanisms that form a balance between HSV latency and replication within ganglia.

Our results have interesting implications for HSV-2 transmission during coitus. Most transmissions occur when an infected person is asymptomatic (*Mertz et al., 1985*, *1998*), a finding that can be interpreted within the context of our model's output. Secondary seeding starts to predictably occur in modeled episodes when viral loads in a single ulcer exceed $10^6$ HSV DNA copies. At this viral load, there are roughly 1000 dead cells, which is not enough for a microulcer to become visible to the naked eye. Therefore, HSV-2 can efficiently seed new regions across the genital tract leading to prolonged high copy number episodes even when shedding is not associated with lesions.

The goal of this study was to design a model that is realistic and adequately complex to recreate our stringently defined dataset, but simple enough to have identifiable parameters. Hence, one

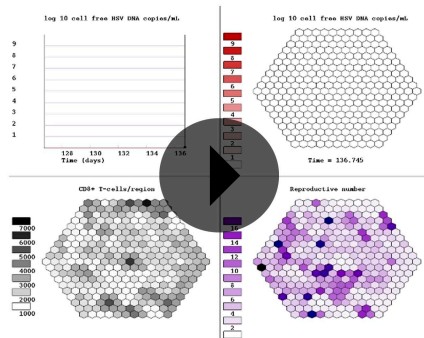

**Video 8**. Spatiotemporal demonstration of immune response to a 14-day episode. The upper left panel represents total cell-free HSV DNA copies per milliliter present over time. The upper right panel represents spatial spread of cell-free virus during the episode. The lower left panel represents CD8+ T-cell density within each region. The lower right panel indicates reproductive number within each region. Quantities are displayed according to a heat map adjacent to each spatial map. The simulated episode is prolonged because virus spreads from the initial region to adjacent regions with low CD8+ T-cell density and reproductive numbers less than one, and is not contained until many regions of the genital tract are infected.

limitation is that the model underestimates immunologic complexity. CD8+ T-cell response is assumed to be a surrogate of the entire immune response. Other key cellular subsets such as CD4+ lymphocytes, natural killer cells, and dendritic cells are not included. Lack of inclusion of innate immunity is another shortcoming as interferon type 1 is present early during lesions (*Spruance et al., 1982*), and contributes to episode containment. Our model generates the hypothesis that chronic HSV-2 infection leads to a dynamic and spatially heterogeneous state of immune readiness. However, the interacting features of host response are not addressed and require further investigation using human and animal models of infection.

For convenience, we divided the genital tract into 300 regions. Simulations with 200–400 regions provided similar fit to the data (not shown). To model spatial spread most completely would require following each cell in the genital tract with an agent-based model as well as accurate spatial elaboration of the genital anatomy (*Bauer et al., 2009*): this would entail great computational cost and inclusion of many unknown parameters. The high number of fitting points in Cohort E and limited number of unknown parameters in our model decreases but does not eliminate the possibility that the model has multiple solutions. If swabs only remove a fixed proportion of free-viral DNA, then it is possible that we are underestimating the genital tract viral load. Based on the complex nature of the observational data, the use of a stochastic simulation model, and the inability to manipulate experimental conditions in our longitudinal studies of human shedding, our parameter values are estimates rather than precise solutions. We believe that the most critical hypotheses generated from this study are qualitative, and are relevant even if certain parameter values were not precisely identified.

In summary, our data suggest intensely localized competition between HSV-2 and the immune system in genital mucosa. Our model raises the possibility that by rapidly spreading between epidermal cells, HSV-2 ensures high viral production within a single ulcer, despite an intense local immune response. Cell-free HSV-2 may prolong episodes via spread to contiguous sites where immune cell density is lower. Each episode is contained by the mucosal immune system, and HSV-2 is often asymptomatic in immunocompetent hosts. Viral strategies that lead to repeated episodes, rapid episode expansion and frequent re-expansion, allow for high shedding frequency, thereby enhancing transmission and disease manifestations. Interventions that target spread of virus between epidermal cells in a single ulcer, and to new regions of the genital tract, will be of value in controlling HSV-2 infection.

## Materials and methods

### Study procedures

Study participants signed informed consent for the human experimentation described in this paper. All potential risks of being involved in the protocols, including discomfort, bleeding and infection, were explained to all participants in detail. The consent process including agreeing to have the experimental data published. Study subjects were reimbursed for clinic visits and the considerable time dedicated to each of the protocols. The University of Washington institutional review board approved all study protocols described in this study.

HSV-2 seropositive participants performed swabs of the genital tract, which were subsequently processed for quantitative HSV DNA PCR using polymerase chain reaction (PCR) (*Wald et al., 1997*; *Magaret et al., 2007*; *Mark et al., 2008*). Detection of HSV DNA was performed using a well-validated collection and detection method (*Magaret et al., 2007*). Swabs of genital secretions were

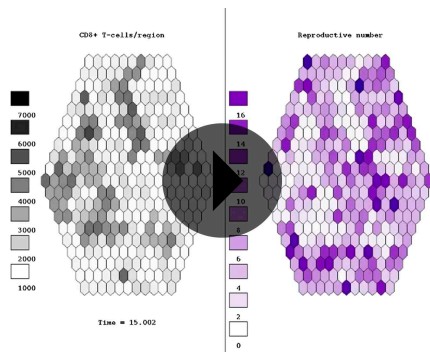

**Video 9**. Spatiotemporal demonstration of immune response over 20 years. The left panel represents CD8+ T-cell density within each region. The right panel indicates reproductive number within each region. Quantities are displayed according to a heat map adjacent to each spatial map. CD8+ T-cells expand rapidly at sites of episodes and then decay slowly over time, correlating with decreases and increases in reproductive number respectively. CD8+ T-cell and reproductive number spatial patterns cycle intermittently between a patchwork of heterogeneous density, broad low density, broad high density, and stark division between regions of high and low density. However, CD8+ T-cell density is virtually always high in at least some regions of the genital tract.

placed into vials containing 1 ml of PCR transport medium and refrigerated until laboratory processing. HSV DNA was detected using a quantitative PCR assay, and was expressed as copies per milliliter of medium. The PCR assay uses type-common primers to the HSV gene encoding glycoprotein B. An internal control was included to ensure that HSV-negative swabs were not due to inhibition. Laboratory personnel were blinded to clinical data.

## Analysis of shedding data

We designed 5 separate cohorts, which used common sample protocols and virological assay but differed according to how frequently patients swabbed their genital tract and according to location of sampling (*Table 1*). Cohort E was used for fitting the model (*Schiffer et al., 2011*): participants swabbed their entire genital tract every 24 hr whether lesions were present or not for a minimum of 30 days. We used eight summary measures (five frequency histograms and three median measures, *Figure 4*), which included a total of 42 data bins, to describe fundamental characteristics of shedding episodes. Three histograms described frequency of different HSV DNA copy number values within logarithmically defined strata (peak, first, and last positive swab within an episode, respectively). Two frequency histograms described episode duration, and frequency of total swabs within logarithmically defined strata. Duration of episodes was measured with interval censoring (whereby episodes of unknown duration due to initiation before day 0 of the shedding protocol or termination following the end of the swabbing period were excluded) and no interval censoring (whereby these episodes were included) (*Figure 4D*). Median measures included regression slopes from initiation to episode peak and episode peak to termination, as well as episode frequency (*Figure 4B,C*).

Cohort C was similar to Cohort E, except participants swabbed every 6 hr (*Mark et al., 2008*), allowing for more precise measures of episode rate, early episode expansion rate, and late episode decay rate. Cohort C was not used for direct model fitting because the number of subjects was much larger in Cohort E. During Cohort C, episodes of age 0–6 hr were captured with the first positive swab value: the median episode within the 2 datasets was captured 3 hr (0.125 days) after shedding episode initiation. We therefore divided median first positive swab copy number by 0.125 to estimate the median early expansion slope. We used an equivalent approach using last positive copy number from episodes to estimate a late decay slope.

We calculated initiation to peak slopes in Cohorts E and C, with the assumption that episodes initiated 12 and 3 hr before the first positive swab, respectively, and fit a regression model from this point to episode peak. We calculated peak to termination slopes in Cohorts E and C, with the assumption that episodes ended 12 and 3 hr after the last positive swab, respectively, and fit a regression model from this episode peak to this point. The narrow interval of swabs from Cohort C was likely to represent a more accurate estimate of expansion and decay kinetics. However, median values from Cohort E (*Figure 4B*) were useful as fitting measures because we also sampled daily when fitting the model.

As detailed in *Table 1*, we performed two shedding protocols (Cohorts A and B) when genital lesions were present. To demonstrate spatial analysis of shedding (Cohort D), study participants had a daily detailed genital examination by an experienced clinician. If lesions were present, the lesion location was recorded on genital diagrams. Twenty-three distinct sites were swabbed at each daily visit (*Figure 2A*). Care was taken to avoid contamination from other anatomic areas during sampling (*Tata et al., 2010*).

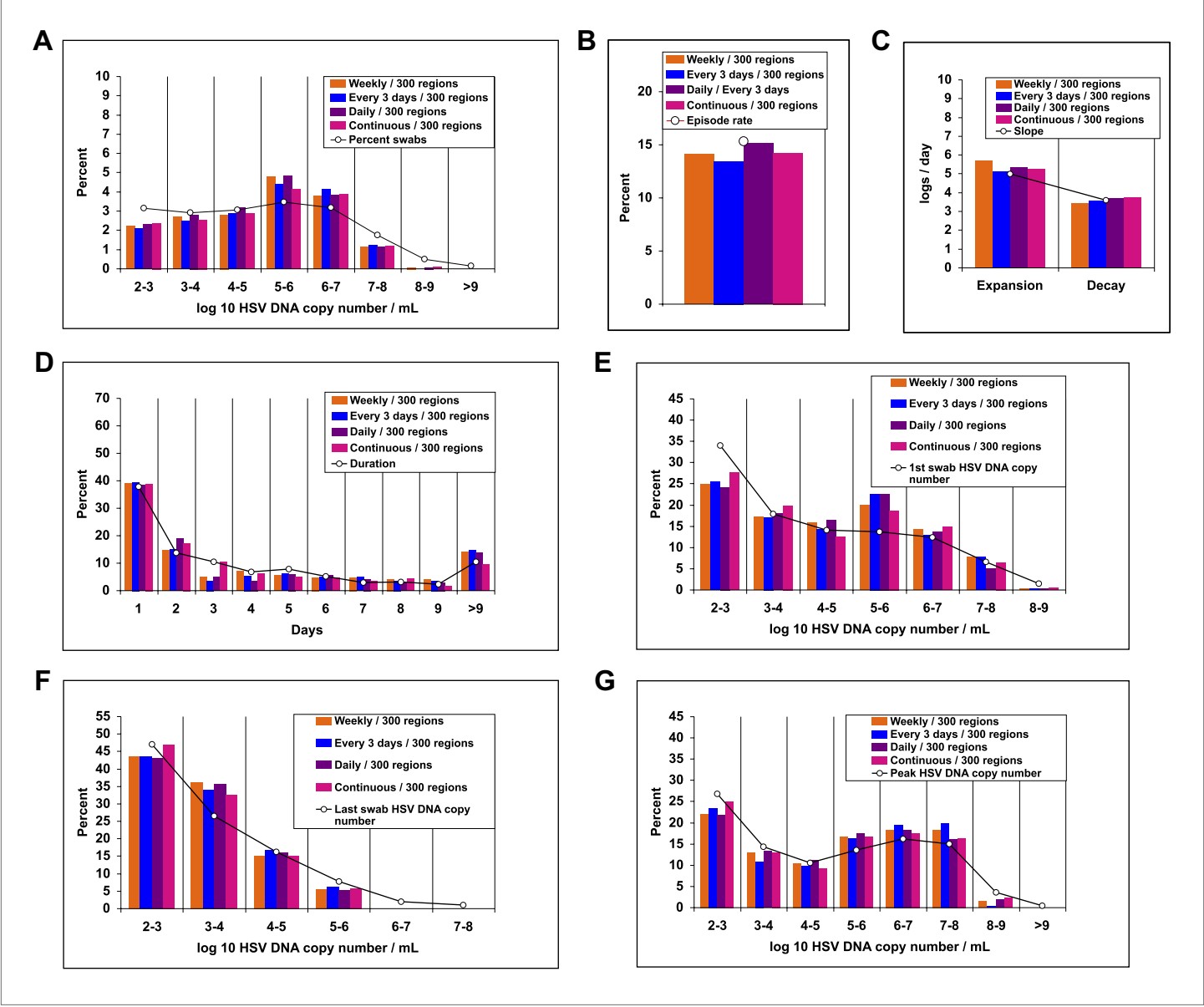

**Figure 6**. Random spatial dispersion of viral particles from neurons reproduced the full diversity of episode characteristics if particles were released continuously, daily, or weekly from neurons. White circles represent results from (**A**) 14,685 genital swabs and (**B**–**G**) 1020 shedding episodes from 531 study participants (**Figure 4**). The model simulations represented with colored bars in each panel continued until 1020 episodes were generated. Sampling occurred every 24 hr as in the clinical protocols. Model output with release of virus randomly throughout the 300 regions on a continuous (pink), daily (purple), every 3 days (blue), and weekly (orange) basis at an average rate of 82 HSV DNA particles per day reproduced (**A**) quantitative shedding frequency and episode, (**B**) rate, (**C**) median initiation to peak slope and peak to termination slopes, (**D**) Duration, (**E**) first HSV DNA copy number, (**F**) last HSV DNA copy number, and (**G**) peak HSV DNA copy number.

The following figure supplements are available for figure 6:

**Figure supplement 1**. Random spatial dispersion of viral particles from neurons reproduced the full diversity of episode characteristics during simulations in which particles were released into only a minority of the 300 modeled regions.

**Figure supplement 2**. Dispersion of viral particles from neurons reproduced the full diversity of episode characteristics during simulations in which particles were released into a minority of modeled regions, provided that dispersion was random rather than clustered within a single geographic region.

**Table 4.** Spatial model simulations that varied only according to duration of sampling (30 days, 60 days, 365 days, and 10 years)

| Simulation duration | | Percent of time with HSV DNA > 150 copies per mL | Percent of time with lesions* present | Episodes per year | Lesions per year |
|---|---|---|---|---|---|
| 30 day | Mean | 13.7 | 7.6 | 11 | 3.2 |
| | Median | 3.4 | 0 | 12 | 0 |
| | Range | 0–82.8 | 0–58.8 | 0–36.5 | 0–24.3 |
| 60 day | Mean | 19.0 | 9.8 | 13.4 | 4.4 |
| | Median | 18.6 | 3.4 | 12 | 6 |
| | Range | 0–54.2 | 0–46.9 | 0–42.6 | 0–18 |
| 365 day | Mean | 19.6 | 10.1 | 14.3 | 4.6 |
| | Median | 19.9 | 9.9 | 14 | 4 |
| | Range | 7.1–36.8 | 2.3–22 | 8–24 | 1–9 |
| 10 year | Mean | 17.5 | 9.1 | 14.5 | 4.3 |
| | Median | 17.6 | 9.0 | 14.5 | 4.2 |
| | Range | 14.8–19.8 | 1.9–12.8 | 11.5–17.1 | 1.4–6 |

*Lesions were defined as > 1 mm diameter ulcers.

Sixty simulations were performed at each of the sampling durations. Within shorter sampling duration simulations, lesion, and shedding frequency varied significantly, while ranges narrowed with prolonged sampling.

## Analysis of biopsy data

We defined spatiotemporal dynamics of HSV-2–specific lymphocyte response to a genital recurrence with in situ staining of genital skin biopsy with HSV-2 antigen-specific quantum dot multimers (*Zhu et al., 2007*, *2009*). We then examined biopsy specimens with confocal microscopy. CD8+ lymphocytes appeared at the precise site of HSV release from the neuron termini, the dermal/ epidermal junction. Previous studies indicated that at least 10% of CD8+ T cells detected in tissue at the dermal–epidermal junction are HSV-2 specific and that this proportion remains constant as CD8+ T-cell levels fluctuate (*Zhu et al., 2007*). Therefore, we used data derived from in situ staining of CD8+ T cells to define relative densities of HSV-2–specific CD8+ T cells in mucosal tissue. In previous studies, absolute number of CD8+ lymphocytes was measured per square millimeter of genital skin. These measures are underestimates for total number of CD8+ T cells present in the entire lesion area because most lesions are >1 mm$^2$. Values of sequential CD8+ T-cell densities gathered every 2 weeks therefore estimate relative intensity of the immune response at different time points.

## Mathematical model fitting to empirical data

All preliminary simulations were performed using Berkley Madonna, while C++ was used for all spatial modeling, high throughput simulations, parameter fitting, sensitivity analyses, and video generation. The goal of our models was to reproduce the 42 data bins within the 8 summary measures of HSV-2 shedding from Cohort E. To this end, we iteratively developed four models with competing equations (*Table 5*). We solved models stochastically due to the random nature of shedding episode initiation and clearance, and to account for the frequent presence of low numbers of infected cells: at each time step, integer values for equation terms were drawn randomly from binomial distributions. To assess the degree of fit between models, numerical output and empirical data required prolonged simulations to minimize fluctuations in output due to stochastic effect. For each fitting attempt, we ran the model until 500–1000 episodes were generated. While state variables were updated at a narrow time interval (0.001 days), we assembled the modeled data exactly as it was gathered in the clinical protocols by sampling every 24 hr.

For each of the four competing mathematical models (*Table 5*), we performed repeated simulations while adjusting and narrowing parameter values (see 'Parameter value search' below) until a weighted least squares approach no longer improved fit to the 8 frequency histograms from Cohort E. For each

**Table 5.** Mathematical models of HSV-2 pathogenesis

| Model | Equations (additions to previous model are denoted in bold) | Variables (model fitting variable) | New features |
|---|---|---|---|
| 1 (*Schiffer et al., 2009*; *Schiffer et al., 2010*) | • $\Delta S = [\lambda - (\beta_i \times S \times V) - (\beta i \times S \times V_{neu})]\Delta t$ <br> • $\Delta I = [(\beta_i \times S \times V_i) + (\beta_i \times S \times V_{neu}) - (a \times I) - (f \times I \times E)]\Delta t$ <br> • $\Delta E = [(F(I) \times \theta \times E) - (\delta \times E)]\Delta t$ <br> • $\Delta V_i = [(p \times I) - (c \times V_i) - (\beta_i \times S \times V_i)]\Delta t$ <br> • $\Delta V_{neu} = [\varphi - (c \times V_{neu}) - (\beta_i \times S \times V_{neu})]\Delta t$ <br> • $F(I) = I/(I + r)$ <br> • $V_{tot} = V_i + V_{neu}$ <br> • $\lambda = d(S - S0)$ | S, I, E, $V_i$ $V_{neu}$, | — |
| 2 | • $\Delta S = [\lambda - (\boldsymbol{\beta i \times S \times V}) - (\boldsymbol{\beta_i \times S \times V_{neu}})]\Delta t$ <br> • $\Delta I = [(\boldsymbol{\beta_i \times S \times V_i}) + (\boldsymbol{\beta_i} \times S \times V_{neu}) - (a \times I) - (f \times I \times E)]\Delta t$ <br> • $\Delta E = [(F(I) \times \theta \times E) - (\delta \times E)]\Delta t$ <br> • $\Delta V_{neu} = [\varphi - (c \times V_{neu}) - (\boldsymbol{\beta_i} \times S \times V_{neu})]\Delta t$ <br> • $\boldsymbol{\Delta V_i = [(p \times I) - (a \times V_i) - (\beta_i \times S \times V_i)]\Delta t}$ <br> • $\boldsymbol{\Delta V_e = [(a \times V_i) - (c \times V_e)]\Delta t}$ <br> • $F(I) = I/(I + r)$ <br> • $\boldsymbol{V_{tot} = V_i + V_e + V_{neu}}$ <br> • $\lambda = d(S - S0)$ | S, I, E, $V_i$, $V_{neu}$, **($V_e$)** | Cell-free and cell-associated particles |
| 3 | • $\Delta S(\boldsymbol{i \ldots 300}) == [\lambda - (\beta_i \times S \times V_i) - (\beta_i \times S \times V_{neu}) - \boldsymbol{(\beta_e \times S \times V_e)}]\Delta t$ <br> • $\Delta I(\boldsymbol{i \ldots 300}) = [\beta_i \times S \times V_i) + (\beta_i \times S \times V_{neu}) - \boldsymbol{(\beta_e \times S \times V_{etot})} - (a \times I) - (f \times I \times E]\Delta t$ <br> • $\Delta S(\boldsymbol{i \ldots 300}) = [(F(I) \times \theta \times E) - (\delta \times E)]\Delta t$ <br> • $\Delta V_{neu}(\boldsymbol{i \ldots 300}) = [\varphi - (c \times V_{neu}) - (\beta_i \times S \times V_{neu})]\Delta t$ <br> • $\Delta V_i(\boldsymbol{i \ldots 300}) = [(p \times I) - (a \times V_i) - (\beta_i \times S \times V_i)\Delta t$ <br> • $\Delta V_e(\boldsymbol{i \ldots 300}) = [(a \times V_i) - (c \times V_e)]\Delta t$ <br> • $F(I) = I/(I + r)$ <br> • $\boldsymbol{I_{tot} = I_1 + I_2 + \ldots + I_{e300}}$ <br> • $\boldsymbol{V_{etot} = V_{e1} + V_{e2} + \ldots + V_{e300}}$ <br> • $\boldsymbol{V_{itot} = Vi_1 + Vi_2 + \ldots + Vi_{300}}$ <br> • $\lambda = d(S - S0)$ | S, I, E, $V_{neu}$, $V_{itot}$, **($V_{etot}$)** | Concurrent plaques from cell-free particles |
| 4* | • $S0 = 1.67e5\,per\,region$ <br> • $\Delta S(\boldsymbol{i \ldots 300}) = [\lambda - (\beta_i \times S \times V_i) - (\beta_i \times S \times V_{neu}) - (\beta_e \times S \times V_e)]\Delta t$ <br> • $\Delta I(\boldsymbol{i \ldots 300}) = [(\beta_i \times S \times V_i) + (\beta_i \times S \times V_{neu}) + (\beta_e \times S \times \boldsymbol{V_{eadj}}) - (a \times I) - (f \times I \times E)]\Delta t$ <br> • $\Delta E(\boldsymbol{i \ldots 300}) = [(F(I) \times \theta \times E) - (\delta \times E)]\Delta t$ <br> • $\Delta V_{neu}(\boldsymbol{i \ldots 300}) = [\varphi - (c \times V_{neu}) - (\beta_i \times S \times V_{neu})]\Delta t$ <br> • $\Delta V_i(\boldsymbol{i \ldots 300}) = [(p \times I) - (a \times V_i) - (\beta_i \times S \times V_i)]\Delta t$ <br> • $\Delta V_e(\boldsymbol{i \ldots 300}) = [(a \times V_i) - (c \times V_e)]\Delta t$ <br> • $\lambda = d(S - S0)$ <br> • $F(I) = I/(I + r)$ <br> • $\boldsymbol{V_{eadj} = V_e}$ **from 6 adjacent regions** <br> • $I_{tot} = I_1 + I_2 + \ldots + I_{300}$ <br> • $V_{etot} = V_{e1} + V_{e2} + \ldots + V_{e300}$ <br> • $V_{itot} = V_{i1} + V_{i2} + \ldots + V_{i300}$ | S, I, E, $V_{neu}$, $V_{itot}$, **($V_{etot}$)** | Spatial model |

*Model 4 has a parameter of regional CD8+ co-dependence ($\rho$) within each plaque-forming region. At episode onset within a region, the CD8+ density is adjusted to infer the spatial co-dependence of CD8 density from surrounding regions: $E_i$ (time + 0.001) = ($E_i \times (1 - \rho)$) + ($E_{avg} \times \rho$) where $E_{avg}$ is average of E from the 6 surrounding regions.

Models are described in the 'Methods'. Units and values of each parameter in the optimized model are listed in *Table 2*. Variables include: S (susceptible skin cells), I (infected skin cells), E (CD8+ T cells), $V_i$ (cell-associated HSV DNA particles), and $V_e$ (cell-free HSV DNA particles).

model fitting simulation, we assigned each of the summary measures (1. episode rate, 2. episode duration, 3. median initiation to peak slope, 4. median peak to termination slope, 5. first positive copy number of episodes, 6. last positive copy number of episodes, 7. peak positive copy number of episodes, and 8. quantitative shedding) a weighting factor to ensure that each summary measure carried an equivalent weight. First, using the empirical data (colored bars, *Figure 4*), the mean value of bins within each of the five histograms (1. episode duration, 2. first positive copy number of episodes, 3. last positive copy number of episodes, 4. peak positive copy number of episodes, and 5. quantitative shedding) was calculated; the inverse square of this value was then used to generate an 'initial weighting factor', which was then divided by the number of bins within the histogram such that each bin was assigned a 'bin weighting factor'. The three median measures (1. episode rate, 2. median initiation to peak slope, and 3. median peak to termination slope) only contained one bin such that the initial weighting factors were equal to the bin weighting factors.

We calculated a 'fitting score' for every simulation of the model. For each bin, the difference between the empirical data and model output was squared and multiplied by the bin weighting factor for the bin, to arrive at a bin score. For the episode duration histogram, if the modeled value fell between the interval censored and noninterval censored empirical values, then the difference was assumed to be zero. Each simulation was given a fitting score equal to the sum of these 42 'bin scores' with a lower score representing better model fit. Using an ad hoc approach, we compared fitting scores with repeated automations of the same model but different parameter sets, to arrive at the optimal set of values for each model; once a model's scores no longer improved, we assigned the model a best fitting score. Best fitting scores were compared between our four models (*Table 6*) in order to select the superior model.

We complemented the best fitting score approach by using 'Akaike information criteria (AIC) scores' as a standardized measure of model agreement with data. The AIC penalizes for extra parameters but rewards better model fit to the data. For our model, the AIC was calculated as $AIC = N \times \ln(RSS/N) + 2k$, where $N$ = number of fitting points, $k$ = number of parameters in the model, and RSS is the residual sum of squares, which we equated to the best fitting score.

To characterize how well each of the four models reproduced individual fitting criteria, we generated a standardized grading system that accounted for the unique nature of our dataset (*Table 6*). For every fitting simulation, we summed the bin scores in each of the eight histograms into individual 'summed criteria scores'. We observed that summed criteria scores >1 resulted in significant disparity between the empirical and modeled data, while scores between 0.5 and 1 signified only moderate differences. Finally, if sums of the scores within a histogram amounted to <0.5, then there was close similarity between the modeled data. For each simulation, if a summed criteria score

**Table 6.** Model fit to Cohort E

| Summary measure | Summed criteria scores | | | | | | | | Best fitting score | AIC |
| | Shedding frequency | Episode duration | First positive swab | Last positive swab | Peak positive swab | Median expansion | Median decay | Episode frequency | | |
|---|---|---|---|---|---|---|---|---|---|---|
| Model 1 | 2.13 | 5.59 | 0.05 | 0.08 | 0.24 | 0.07 | 0.43 | 0.01 | 8.61 | −50 |
| Model 2 | 1.26 | 8.33 | 0.38 | 0.34 | 0.22 | 0.04 | 0.09 | 0.03 | 10.69 | −41 |
| Model 3 | 0.25 | 3.75 | 0.31 | 0.22 | 0.61 | 0.05 | 0.50 | 0.22 | 5.92 | −62 |
| Model 4 solved for 10 parameters | 0.25 | 0.13 | 0.29 | 0.15 | 0.09 | 0.01 | 0.01 | 0.03 | 0.96 | −139 |
| Model 4 solved for 5 parameters | 0.39 | 0.32 | 0.44 | 0.21 | 0.09 | 0.04 | 0 | 0.18 | 1.67 | −125 |

Summed criteria scores measure the degree of fit for each model according the eight individual shedding episode features using a weighted sum of squares. Model 4 is the spatial model. Models 1–3 are described in the 'Methods'. Best fitting score is a sum of all summed criteria scores for a particular model with lower scores indicating better fit. AIC: Akaike information criteria with lower scores indicating better fit.

exceeded 1, then we labeled the simulation a 'poor' fit to the summary measure. If a summed criteria score exceeded 5, then we labeled the simulation a 'very poor' fit. If a summed criteria score exceeded 0.5 but was <1.0, we labeled the simulation a 'fair' fit, while scores <0.5 were considered a 'good' fit. Using this approach, we specified which facets of shedding certain models could not replicate. We used summed criteria scores to provide ranges for optimized parameter values (**Table 2**).

## Mathematical model 1: baseline model of viral replication and CD8+ T-cell response

The initial model that we tested in our analysis (Model 1, **Table 5**) has been described (**Schiffer et al., 2009**, **2010**). The HSV-2 replication cycle in the genital tract and immunologic response to HSV-2–infected cells can be referenced to each of the model's parameters. The model assumes chronic infection. During the 'latent phase' of chronic genital HSV infection, virus is maintained in a dormant state in the sacral ganglia. During 'reactivation', virions are released from neuronal endings at the dermal–epidermal junction at a certain rate (ϕ). Released HSV-2 survives in the genital tract for a defined duration (1/$c$), during which time it can transfer from neurons and infect local epithelial cells. Viral infectivity (1/[β$_i$ × S]) is roughly defined as the number of viruses that are needed to infect one epithelial cell per day assuming the constant presence of viruses and a full complement of susceptible cells.

For all of the models in this manuscript, we assumed some degree of target cell limitation, by virtue of susceptible cells decreasing relative to total number of cells with time. In one sense, target cells appear to be limitless in the genital tract: visualization of skin during lesions reveals a high ratio of uninfected to infected keratinocytes in genital skin (**Figure 2C**); even in severely immunocompromised hosts with severe infections, ulcers never involve the entire genital tract and do not spread more than a few centimeters down the thigh or up the stomach or back. Yet, spread of virus from the cytosol of one cell to another is constrained by the number of cellular contacts of an infected cell: as an ulcer spreads in radius, an increasing proportion cells in contact with an infected cell are already infected. This effect may decrease average infectivity of an infected cell as ulcer radius expands. Therefore, the infectivity term does decrease with time in our model according to number of cells that remain susceptible.

We assumed that if an epithelial cell becomes infected, then it will die via direct lysis if it survives long enough to become packed with viruses (lifespan = 1/$a$), or via CD8+ lymphocyte–mediated killing (lifespan = 1/[$f$ × E]). Lymphocyte killing efficiency ($f$) is defined as the number of infected cells cleared by one CD8+ lymphocyte cell per day in vivo. If an epithelial cell evades CD8+ lymphocyte mediated killing for the entire duration of cellular infection, it will produce a total of $p/a$ viruses: $p$ is the rate of viruses produced by an infected cell per day. For each fitting simulation, we included a minimum value of E within each region to avoid viral loads >$10^{10}$ HSV DNA copies, which are never present clinically.

HSV-2 reactivation that results in mucosal HSV shedding can be subclinical or clinical. Clinical shedding is termed a 'recurrence' and is associated with sporadic crops of vesicles, which form due to the death of closely congregated epithelial cells. Vesicles progress to ulcers and then undergo healing due to regrowth of susceptible cells: we assume that regrowth occurs according to a growth rate, λ or $d$ × (S0 − S) with growth limited by S0, the carrying capacity of the system. 'Subclinical shedding episodes' are common during chronic HSV-2 infection and are defined by local detection of virus either by culture or PCR, in the absence of a clinically apparent genital lesion or symptoms (**Mark et al., 2008**).

The formation of a genital lesion is accompanied by rapid accumulation of localized CD8+ lymphocytes at the dermal–epidermal junction at a peak rate (θ), followed by slow decay of these cells over a period of months after lesion healing (lifespan = 1/δ). The CD8+ replication rate in our model is saturated at θ, and θ/2 occurs when infected cells are equal in number to parameter $r$, which represents how many epithelial cells need to be infected before half-maximal CD8+ expansion. Therefore, parameter $r$ defines how rapidly immune cells recognize and respond to viral antigens on the surface of infected cells. We have previously demonstrated that local replication of CD8+ T cells rather than trafficking from other sites is the most mathematically likely means of viral control (**Schiffer et al., 2009**, **2010**).

Model 1 recreated a 2-cm-diameter circular area (surface area = 314 mm$^2$) of susceptible epithelial cells in the anogenital mucosa as the typical maximum size for a single ulcer. We chose this diameter because the average surface area of recurrent genital HSV lesions (which may include many ulcers) in immunocompetent persons is 60 mm$^2$ (range 2–270 mm$^2$) (**Corey et al., 1983**), which equates to

a lesion of diameter 8.7 mm (range 1.6–18.5 mm). Ulcers are rarely >3 mm in diameter. Thus, we designed the model to be inclusive of the vast majority of recurrent genital ulcers in immunocompetent persons. Model 1 failed to create episodes with multiple peaks, and failed to produce episodes of >4 days duration (~33% of clinical episodes) (*Table 6*).

## Mathematical model 2: differentiation of cell-associated and cell-free viruses

In an attempt to improve model fit, we iteratively updated the model to more realistically represent infection, according to our clinical and experimental observations of HSV-2 biology. We next constructed a model, which differentiated cell-associated from cell-free particles (Model 2) and assumed that cell-associated particles ($V_i$) can passage from cell to cell as soon as they form within a cell: this assumption is based on studies that document transfer of HSV-2 particles via tight junctions that bind epidermal cells to one another (*Collins and Johnson, 2003*). Model 1 made the assumption that all viral particles had this property because if we assumed that virus only became infectious after infected cell lysis, then rate of episode expansion was far too slow. (In fact, the majority of published virological models to date, by assuming homogeneous mixing, also assume that newly formed particles are instantly infectious, which implies cell-to-cell spread.)

In Model 2, cell-associated particles ($V_i$) converted to cell-free virus ($V_e$) after infected cells ruptured. We assumed in Model 2, that cell-free particles were noninfectious and decayed according to a constant linear function, whereas cell-associated particles decayed according to the decay rate of infected cells, which in turn is a function of infected cell death rate and CD8+ T-cell density. Because swabs probably do not remove most cell-associated particles and infected cells (otherwise swabbing would mitigate shedding), we fit $V_e$ to the data in Cohort E. Model 2 also failed to create episodes with multiple peaks, and failed to produce episodes of >4 days duration (*Table 6*).

## Mathematical model 3: a nonspatially constrained model with numerous regions of concurrent HSV-2 replication

Based on clinical observations outlined in *Figure 2*, our next models were designed to test a hypothesis that multiple concurrent plaques may explain prolonged episodes due to re-expansion. Model 3 consisted of 300 possible regions of infection, linked by the ability of neurons to randomly release virus into any of the 300 regions. Moreover, we allowed formation of new plaques within other regions by cell-free particles. Cell-free particles in the model were assigned a new infectivity parameter ($\beta_e$). In this model, there was no spatial constraint of virus, such that HSV from one region could initiate new sites of replication in the 299 other regions. We included parameter ($\epsilon$) to account for delay in viral production within secondary plaques after seeding. This parameter was estimated based on simple experiments that we conducted with a red fluorescent protein-VP26 HSV construct that turns cells red following viral replication and encapsidation of viral DNA (VP 26 is a late capsid protein). Individual cells turn red approximately 12–24 hr after very low multiplicity of infection (MOI = 0.001) of Vero cells, which are a cell line of monkey kidney epithelial cells that are permissive for HSV entry, replication, and spread. Therefore, $\epsilon$ was assigned an initial value range of 0.5–1. Model 3 produced longer episodes in general than Model 2, and successfully created episodes with re-expansion. Nevertheless, episodes were too short in duration to reproduce the frequency histogram for episode duration (*Table 6*).

## Mathematical model 4: the spatial model

Model 4 (*Figure 3*) is described in the 'Results' and *Table 5*. To slow spread from region to region and because HSV-2 plaques often congregate in proximity to one another, we created a spatial diagram of the genital tract and limited spread of cell-free particles from infected regions to contiguous regions only. Regions were arrayed in a two-dimensional matrix of hexagons such that cell-free particles from a region could only infect a maximum of six other regions (*Figure 3*), thus limiting rate of HSV-2 spread from region to region within the genital tract. For edge regions within Model 4, there were only 4 contiguous regions at risk.

The regions were linked immunologically according to parameter ρ. When an ulcer was initiated in a region (infection of a single epidermal cell), the CD8+ T-cell density was assumed to include a certain proportion of CD8+ T-cell density in surrounding regions. If ρ = 0, then CD8+ T-cell density in a region was independent of surrounding regions. This implies that all immunity was localized only to an area with viral replication. If ρ = 1, CD8+ T-cell density in a region was assumed to equal the average from surrounding regions.

## Parameter value search

We explored >1000 parameter sets within each of our four possible models using Latin hypercube sampling. The goal was to arrive at a parameter value set that maximized the fit of each model to the shedding summary measures. Each parameter was initially assigned a wide possible range of values based on previous literature review (*Schiffer et al., 2009*). The typical search scheme involved adjusting three randomly selected parameters at a time with five possible parameter values for each parameter leading to $5^3$ or 125 parameter sets: the parameter set with the closest fit to model output was selected for the next round of simulations. There were a maximum of 10 unknown parameters per model so that one cycle through the entire parameter set involved 375–500 simulations (3–4 rounds of 125 simulations). After a single cycle through all parameter sets, several more cycles were performed with parameter ranges re-expanded around revised best parameter values. When there was no improvement with successive cycles through the parameter sets, we narrowed the ranges by 50% and cycled through the 375- to 500-parameter sets again. The three parameters that were concurrently varied were randomly selected with each fit. For each model, we continued this process of parameter set selection and parameter range narrowing until the fitting scores no longer improved. The best fitting score for a particular model was then used for comparison to other models.

Because of the stochastic nature of our model, there was a reasonable amount of run-to-run variability, even with 10-year simulations (*Table 4*). We therefore repeated simulations with our spatial model that ultimately reproduced the entirety of our dataset. We arrived at similar parameter values when repeating the fitting process.

Five of the 10 model parameters used in the spatial model (Model 4) have been estimated experimentally (*Schiffer et al., 2009*). These values included $\beta_i$ = 6e-8 DNA copy days per cell (~100 viruses needed per day to infect one adjacent cell) (*Schiffer et al., 2009*), $p$ = 1e5 HSV DNA particles per day (based on ~1e3 particle forming unit [pfu] per cell and ~100 HSV DNA copies per pfu) (*Sacks and Schaffer, 1987*; *Jiang et al., 2007a*; *Jiang et al., 2007b*), $c$ = 10 per day (*Schiffer et al., 2009*; *Turner et al., 1982*) and $\theta$ = 2.9 per day (*De Boer et al., 2003*). A pfu represents a viral particle that can form a new infectious plaque in cell culture. Pfus are produced at a lower rate than HSV DNA because not all replicated viral genomes are infectious. As above, we assigned a value for $\varepsilon$ = 0.66. We therefore attempted model fit by varying only the remaining five parameters to minimize the likelihood of multiple solutions to our model. We tested four parameter values in this case leading to $4^5$ or 1024 parameter sets: the parameter set with the closest fit to model output was selected for the next round of simulations. When fitting with 5 rather than 10 unknown parameters, the sum of squares and AIC scores indicated a very slight decrease in fit to Cohort A data (*Table 6*).

## Model output analysis

A feature shared by all models described in this manuscript is sensitivity to CD8+ T-cell density at episode onset, which had an important effect on amount of virus produced per episode (*Schiffer et al., 2010*). We started each simulation with zero infected cells and viruses. To avoid creating a bias on model outcomes due to initial CD8+ T-cell values, we ran the model with its particular parameter set for 365 days, and then used the CD8+ T-cell values in each region at the end of these 365 days as the starting values for the recorded simulation. For analyses of shorter duration (e.g., 30- to 60-day simulations in *Table 4*), we ensured a random distribution of CD8+ T-cell starting values by randomly selecting a starting time between 4 and 10 years for sampling after initial simulations of the model.

We calculated the reproductive number ($R = [p \times \beta_i \times S0]/([a + fE] \times c$) continually during each simulation. The reproductive number is the average number of cells that an infected cell would infect assuming the presence of CD8+ T cells. For our spatial models, which divided the genital tract into 300 separate regions that were susceptible to viral replication, we calculated R separately within each region. We calculated plaque diameter in millimeters (diameter = $8.32e-3 \times \sqrt{(S0 - S - I)}$) continually during each simulation by estimating the number of missing cells due to cell death from infection and using the diameters of these cells to estimate the size of each plaque. Our formula for plaque diameter incorporates measures of individual cell volume and ulcer depth: derivations of our estimates for R and plaque diameter are available in previous publications (*Schiffer et al., 2009*, *2010*). Our formula for diameter predicts that the loss of 14,400, 57,800, and 130,000 cells at any given point in time will lead to 1-, 2-, and 3-mm-diameter ulcers, respectively.

Upon arriving at the most accurate model and set of parameter values, we conducted numerous simulations, which are described in the 'Results', to better understand the model's dynamic features.

We simulated 100 episodes with at least 10,000 infected cells within a single ulcer-forming region to describe features of more severe episodes. Next, we calculated the infected cell growth rate ($g$) from 1 to peak cells for each episode ($g = (\ln I_{peak})/t_{peak}$), and used this measure to roughly estimate generation time ($T_g$, time from becoming infected to infecting a neighboring cell for an infected cell), where $T_g = ((R_{init} − 1)/2))/g$ (**Fraser et al., 2004**). The reproductive number was halved in this equation because the number transitions from $R_{init}$ to a value of 1 at episode peak. We similarly calculated infected cell doubling time for each episode according to: $\ln (2) T_g/((R_{init} − 1)/2)$.

## Sensitivity analyses

Our local sensitivity analysis was based on conducting prolonged simulations with single parameters adjusted until fit to the dataset was no longer optimal. We also performed global sensitivity analyses with Monte Carlo simulations to better understand critical dynamics of the system.

For global sensitivity analyses, all model parameters were randomly drawn from probability distribution functions (pdfs), which were derived according to ranges of parameters from model fit (**Table 2**). We first simulated 500 episodes each of which was assigned a unique parameter set. Region and timing of episode initiation was selected stochastically to allow a range of initial CD8+ T-cell densities. Partial rank correlation coefficients (PRCCs) (**Marino et al., 2008**) and Akaike Information Criteria (AIC) were used to identify important predictive parameters of episode duration and peak viral load. We next performed 500 10-year simulations each associated with a unique parameter set, and examined the effect of parameter variability on shedding frequency and episode rate.

## Acknowledgements

We express gratitude to our study participants, to Varsha Dhankani and Laura Matrajt for careful review of the manuscript, and to Mindy Miner for technical editing.

# Additional information

### Funding

| Funder | Grant reference number | Author |
| --- | --- | --- |
| National Institutes of Health | P01 AI030731 | Joshua T Schiffer, Amalia Magaret, Christine Johnston, Karen E Mark, Jia Zhu, Meei-Li Huang, Anna Wald, Lawrence Corey |
| National Institutes of Health | R37 AI042528 | Jia Zhu, Lawrence Corey |
| National Institutes of Health | K24 AI07113 | Anna Wald |
| National Institutes of Health | K23 AI087206 | Joshua T Schiffer |
| National Institutes of Health | K23 AI079394 | Christine Johnston |
| National Institutes of Health | K23 AI071257 | Karen E Mark |
| National Center For Advancing Translational Sciences of the National Institutes of Health | UL1TR000423 | Joshua T Schiffer |

The funders had no role in study design, data collection and interpretation, or the decision to submit the work for publication.

### Author contributions

JTS, Conception and design, Acquisition of data, Analysis and interpretation of data, Drafting or revising the article, Contributed unpublished essential data or reagents; DS, RAS, Conception and design, Analysis and interpretation of data; AM, Analysis and interpretation of data, Drafting or revising the article; CJ, Conception and design, Acquisition of data, Analysis and interpretation of data, Drafting or revising the article; KEM, SS, NO, SK, Conception and design, Acquisition of data; JZ, BR, M-LH, Acquisition of data, Analysis and interpretation of data; KRJ, AW, LC, Conception and design, Drafting or revising the article

## Ethics

Human subjects: Study participants signed informed consent for the human experimentation described in this paper. All potential risks of being involved in the protocols, including discomfort, bleeding and infection, were explained to all participants in detail. The consent process included agreeing to have the experimental data published. Study subjects were reimbursed for clinic visits and the considerable time dedicated to each of the protocols. The University of Washington institutional review board approved all study protocols described in this study (IRB approval numbers 37460 and 38090). All clinical investigations were conducted according to the principles expressed in the Declaration of Helsinki.

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
