## [Decision Letter]

Thank you for sending your work entitled “Rapid Localized Spread and Immunologic Containment Define Herpes Simplex Virus-2 Reactivation in the Human Genital Tract” for consideration at *eLife*. Your article has been evaluated by a Senior editor and three reviewers, one of whom is a member of our Board of Reviewing Editors.

The editors and the reviewers discussed their comments before we reached this decision, and the editors have assembled the following comments to help you prepare a revised submission.

1) The paper lacks a proper sensitivity analysis that correlates parameter variability to outcomes of interest. This really needs to be performed to make the results complete. While the parametrization is robust and identifiability does not seem to be a major issue in this case, a sensitivity analysis could give additional insight in the dynamics of infection.

In the manuscript, two very concise statements are given on the topic from which one understands that sensitivity analysis is only used to identify the range of parameter values by progressively modifying one at a time the values of parameters and keeping all other fixed (to which value?) until the fitting score exceeds a quality threshold. However, a multi-dimensional parameter space exploration is more suited to understand the relationships between input and model's output and can allow the identification of critical dynamics of the system (see, for example, Marino et al., JTB, 2008).

2) What does the model suggest are likely drivers of variability between patients?

3) How sensitive is the model to seeding frequency or location?

4) What is the impact of the starting distribution of immune cells on the dynamics or success of lesions?

5) What is the main driver of the size of the lesions?

6) Please discuss the likely implications of the work for transmission and how it is related to the presence or absence of lesions.

---

## [Author Response]

*1) The paper lacks a proper sensitivity analysis that correlates parameter variability to outcomes of interest. This really needs to be performed to make the results complete. While the parametrization is robust and identifiability does not seem to be a major issue in this case, a sensitivity analysis could give additional insight in the dynamics of infection*.

*In the manuscript, two very concise statements are given on the topic from which one understands that sensitivity analysis is only used to identify the range of parameter values by progressively modifying one at a time the values of parameters and keeping all other fixed (to which value?) until the fitting score exceeds a quality threshold. However, a multi-dimensional parameter space exploration is more suited to understand the relationships between input and model's output and can allow the identification of critical dynamics of the system (see, for example, Marino et al., JTB, 2008)*.

We agree that a multi-dimensional parameter space search is a highly informative method to interrogate the true dynamics of the model and to characterize which parameters may drive key model outcomes. In the revised version, we include such an analysis. We performed Monte Carlo simulations in which parameter values were drawn randomly from probability distribution functions. Akaike information criteria and partial rank correlation coefficients were employed to correlate model inputs with outcomes that were selected based on clinical and biologic relevance. The Marino article was extremely helpful in guiding this analysis and is added as a reference (Marino, 2008).

We performed two sets of analyses: first, we examined the effect of parameter variability on two short-term model outcomes: duration and peak viral production of 500 simulated episodes; second, a global sensitivity analysis was performed to examine more long-term shedding outcomes measured in 500 10-year simulations. Selected long-term outcomes included shedding rate and episode rate.

*2) What does the model suggest are likely drivers of variability between patients*?

Using the global sensitivity analysis, we explored possible drivers of inter-personal variability. First, we explored the short-term effects of model parameter values on shedding episode characteristics. As in our past models, CD8+ T-cell density at the initial reactivation site was the critical driver of peak episode viral load and duration of the episode (Schiffer et al., 2011; Schiffer et al., 2011). However, the fact that this value was not completely predictive highlights the importance of stochastic effects on short-term viral trends in our spatial model. Increased viral replication rate in epidermal cells modestly enhanced episode duration and viral production, while increased viral infectivity modestly prolonged episode duration. These results are summarized in the Results section and Table 3, and their biologic significance is reviewed in an updated Discussion.

Predictors of long-term shedding variability between patients are also summarized in Table 3, as well as a new section at the end of the Results. In the 500 10-year Monte Carlo simulations, no single model parameter pertaining to mucosal biology was completely predictive of shedding rate or episode rate. Surprisingly, higher CD8+ T-cell expansion rates correlated with higher shedding rate, while lower epidermal production rate of HSV-2, perhaps mediated by increased force of innate immunity, also correlated with higher shedding rate.

These counterintuitive findings can be explained in terms of the model's complex spatial predator-prey dynamics between immune cells and productively infected cells. Under simulated conditions favoring lower capacity for short-term viral production in mucosa (lower viral replication rate in epidermal cells or higher CD8+ replication rate), a higher proportion of simulated episodes were rapidly contained in less than a day: these episodes typically involved <10 infected cells and were cleared in <24 hr, prior to the need for T-cell reconstitution. As a result of this general pattern, CD8+ T-cells replenished less frequently and the typical duration between episodes of >24 hr increased. Due to more prolonged phases of CD8+ T-cell decay, the average CD8+ T-cell density at the onset of severe episodes was lower. Therefore, under conditions of sustained lower viral replication rate in epidermal cells and/or higher CD8+ replication rate, while simulated episodes of <1 day were more common, episodes of >10 days were also more common: parameter conditions that seemingly would promote less shedding in fact predicted slightly higher overall shedding rates. This more prolonged decay phase of CD8+ T-cells between episodes may be considered a form of T-cell exhaustion.

Lower viral replication rate in epidermal cells and higher CD8+ replication rate also correlated with higher measured episode rate. This finding may represent an artifact of overlapping shedding episode initiations within different spatial regions. In fact, neither of these parameters has any effect on the rate of shedding episode *initiation* within mucosa: shedding episode initiation rate is determined only by the rate of release of HSV-2 from neurons as well as viral infectivity. Rather, under these low replication conditions, episodes may be less likely to initiate when another episode is already ongoing. Therefore, a higher number of episode initiations are detected rather than being masked by an ongoing reactivation. These predictions highlight the incredible complexity of virus host interplay within mucosa when spatial features of pathogenesis are considered.

While these findings are intriguing, in a revised version of the Discussion, we highlight that the global sensitivity analyses are exploratory in nature. First, feasible probability density functions for many of the model's parameters can only be estimated. In addition, it is unknown whether our model's parameters are independent or collinear. For instance, some investigators believe that CD8+ T-cells are vital for controlling the kinetics of HSV-2 release within ganglia (Khanna, 2003). Therefore, viral release rate from neurons and CD8+ expansion rate in periphery may be co-dependent. There is no available data to support or refute this idea. We therefore emphasize that sensitivity analysis results are hypothesis generating and raise several fruitful avenues for future research.

*3) How sensitive is the model to seeding frequency or location*?

The global sensitivity analysis also demonstrated that quantitative shedding rate and episode rate correlate with average number of HSV-2 DNA genomic copies released into the genital tract from neurons per day. Regarding spatial seeding, we now include analyses that demonstrate that if viral release was heterogeneously disseminated across the genital tract, but was limited to <10% of selected model regions, then model fit started to break down with over representation of short duration episodes with low peak copy numbers. If viral release from neurons was allowed in a higher percentage of micro-regions, but was limited to one geographic sector of the genital tract, then model fit was less robust as well. We conclude based on these analyses, as well as data in Figure 2a, which shows brief HSV-2 reactivations in spatially distinct regions, that HSV release from neurons is likely to consist of highly frequent and widely spatially dispersed release of small amounts of viral genomes. This viral strategy allows regions with low CD8+ densities and high viral growth potential to be periodically seeded, promoting larger outbreaks.

In separate analyses, we identified that similar shedding characteristics occur whether the virus is dripped continually or pulsed into single regions on a daily or weekly basis, provided that the total number of viruses per day is the same. This finding is in keeping with findings from a prior publication from our group (Schiffer et al., 2011).

The above findings are represented with Figure 6 & Figure 6–figure supplements 1 & 2, which highlight in detail the effect of changes in seeding frequency and location on episode dynamics.

*4) What is the impact of the starting distribution of immune cells on the dynamics or success of lesions*?

As discussed above, we now include a global sensitivity analysis in which the effect of parameter variability on a single episode is assessed. For each of these 500 episodes, the spatial array of CD8+ T-cell densities was allowed to develop naturally, the model simulation was started at a random time, and the episode was initiated stochastically in one of the 300 regions. For each episode, the CD8+ T-cell density in the initiating region was evaluated as a possible exposure variable for episode duration and peak copy number. Moreover, parameter values of viral replication and immune response were chosen randomly from the probability density functions described above.

We identified that CD8+ T-cell density in the initial area and surrounding areas was more predictive of episode severity than model parameter variability.

*5) What is the main driver of the size of the lesions*?

When considered in aggregate, the above analyses suggest that spatial immune density at the precise site of a genital reactivation has the most immediate predictive effect on shedding episode severity and lesion size, while a complex intersection of immune and virologic parameters in the model are likely to dictate long-term shedding patterns over months and years.

*6) Please discuss the likely implications of the work for transmission and how it is related to the presence or absence of lesions*.

We include a paragraph in the Discussion with our thoughts on how this work is relevant to HSV-2 transmission dynamics. Coital transmission can occur when shedding is either symptomatic or asymptomatic. In model simulations, seeding of adjacent regions can occur when the viral load in a single region is ∼10^6^ HSV DNA copies, which corresponds with a number of killed cells that is too low to become visible to the human eye. This explains why episodes with reasonably high viral loads and prolonged duration, but no visible lesions, are quite common. We conclude that the phenomenon of secondary seeding is important for prolonging both symptomatic and asymptomatic episodes, and is therefore likely to enhance transmission under both circumstances.

References

1. Marino S, Hogue IB, Ray CJ, Kirschner DE. 2008. A methodology for performing global uncertainty and sensitivity analysis in systems biology. *J Theor Biol*
**254**:178–96.

2. Schiffer JT, Abu-Raddad L, Mark K, Zhu J, Selke S, et al. 2009. Frequent Release of Low Amounts of Herpes Simplex Virus From Neurons: Results of a Mathematical Model. *Science Translational Medicine*
**1**:1–9.

3. Schiffer JT, Abu-Raddad L, Mark KE, Zhu J, Selke S, et al. 2010. Mucosal host immune response predicts the severity and duration of herpes simplex virus-2 genital tract shedding episodes. *Proc Natl Acad Sci USA*
**107**:18973–8.

4. Khanna KM, Bonneau RH, Kinchington PR, Hendricks RL. 2003. Herpes simplex virus-specific memory CD8+ T cells are selectively activated and retained in latently infected sensory ganglia. *Immunity*
**18**:593–603.